# HTRA1 disaggregates α-synuclein amyloid fibrils and converts them into non-toxic and seeding incompetent species

Sheng Chen[1], Anuradhika Puri [1], Braxton Bell[1], Joseph Fritsche[1], Hector H. Palacios[1], Maurie Balch[1], Macy L. Sprunger[1], Matthew K. Howard [1], Jeremy J. Ryan [1], Jessica N. Haines[2], Gary J. Patti [1,3], Albert A. Davis [2] & Meredith E. Jackrel [1] ✉

Parkinson's disease (PD) is closely linked to α-synuclein (α-syn) misfolding and accumulation in Lewy bodies. The PDZ serine protease HTRA1 degrades fibrillar tau, which is associated with Alzheimer's disease, and inactivating mutations to mitochondrial HTRA2 are implicated in PD. Here, we report that HTRA1 inhibits aggregation of α-syn as well as FUS and TDP-43, which are implicated in amyotrophic lateral sclerosis (ALS) and frontotemporal dementia. The protease domain of HTRA1 is necessary and sufficient for inhibiting aggregation, yet this activity is proteolytically-independent. Further, HTRA1 disaggregates preformed α-syn fibrils, rendering them incapable of seeding aggregation of endogenous α-syn, while reducing HTRA1 expression promotes α-syn seeding. HTRA1 remodels α-syn fibrils by targeting the NAC domain, the key domain catalyzing α-syn amyloidogenesis. Finally, HTRA1 detoxifies α-syn fibrils and prevents formation of hyperphosphorylated α-syn accumulations in primary neurons. Our findings suggest that HTRA1 may be a therapeutic target for a range of neurodegenerative disorders.

Protein misfolding is associated with multiple neurodegenerative disorders including Parkinson's disease (PD), amyotrophic lateral sclerosis (ALS), and frontotemporal dementia (FTD)[1]. α-Synuclein (α-syn) is an abundant neuronal protein with several putative roles, including modulation of synaptic transmission[2]. In PD and other synucleinopathies, α-syn undergoes a structural conversion from its native soluble state into β-sheet rich amyloid fibrils[1,3]. The accumulation of α-syn in cytoplasmic Lewy bodies is the pathological hallmark of PD[1]. The misfolding of α-syn may lead to its inactivation and loss of native function. Additionally, accumulation of misfolded α-syn confers a toxic gain of function[4,5]. α-Syn amyloid fibrils are highly insoluble and resistant to proteases and other denaturants[6]. Additionally, α-syn fibrils can enter neighboring cells and seed further misfolding of monomeric α-syn[2,4]. In ALS and FTD, several proteins with prion-like domains can misfold, undergo aberrant phase transitions, and

aggregate[7,8]. These proteins include TDP-43 and FUS, both of which mislocalize from the nucleus to the cytoplasm where they aggregate and are associated with both a toxic gain of function of the misfolded species along with a loss of function due to their sequestration in the cytoplasm[7]. A better understanding of the molecular mechanisms by which cells avert formation of amyloid and other aggregated species, as well as how they might clear misfolded species to avoid further aggregation, is essential for the ultimate development of new therapeutic strategies. Recently, there have been major advances in the development of therapeutics for neurodegenerative disorders. Specifically, drug development targeting amyloid plaques in Alzheimer's disease has progressed significantly, leading to the FDA-approval of therapeutics including Aducanumab[9] and Lecanemab[10]. These drugs provide promising proof-of-concept that targeting amyloid species can modify disease progression. However these new

---

[1]Department of Chemistry, Washington University, St. Louis, MO 63130, USA. [2]Department of Neurology, Washington University, St. Louis, MO 63130, USA. [3]Department of Medicine, Washington University, St. Louis, MO 63130, USA. ✉e-mail: mjackrel@wustl.edu

therapeutics have limited efficacy, slowing, but not reversing, disease progression[11].

To preserve protein homeostasis (proteostasis), protein quality control systems have evolved to promote the proper folding of proteins, as well as to repair and degrade proteins when necessary[12]. However, a range of different proteins can adopt a misfolded and insoluble β-sheet amyloid secondary structure which can preclude their clearance by the proteostasis network[8]. Through activation of the heat shock response, chaperone proteins can be upregulated and recruited to aggregates and amyloid. However, chaperones are only able to prevent further aggregation, and are often insufficient to solubilize or clear these accumulations[13]. In contrast, protein disaggregases are capable of engaging and dissolving otherwise insoluble amyloid fibrils, pre-amyloid oligomers, and other aggregates[14,15]. The yeast disaggregase Hsp104 has been demonstrated to eliminate fibrils of not only yeast prions, but also proteins associated with PD, Alzheimer's disease (AD), Huntington disease (HD), and other disorders[16–18]. While humans do not express Hsp104, human disaggregases have been identified, including Hsp110[19–21], VCP[22], DAXX[23], Kapβ2[24], and TRIM11[25].

High-temperature requirement A (HTRA) proteins are ATP-independent PDZ serine proteases[26]. HTRA proteases are conserved and found in bacteria, fungi, plants, and animals, with many organisms expressing more than one HTRA isoform in different cellular compartments[27]. HTRA proteins are known to function in the stress response, whereby they are thought to bind damaged proteins via their PDZ domain and mediate proteolysis[28,29]. HTRA1 proteolytic activity is activated by inter-monomer communication that requires trimeric assembly. Activity can be regulated through allosteric activation mediated by PDZ domain binding[30,31]. PDZ domains are known to mediate binding to substrates harboring β-sheets via a β-sheet augmentation mechanism, whereby the β-sheet rich region of the PDZ domain binds the β-sheet of the substrate[27,32]. Such a mechanism might also promote binding to β-sheet rich amyloid species. Indeed, HTRA1 can disintegrate and proteolyze fibrillar tau aggregates[28]. HTRA1 is a ubiquitously expressed protein that is secreted into the extracellular matrix[27]. Intracellularly, HTRA1 is found in the cytoplasm, associated with microtubules, and in the nucleus. HTRA2 is expressed in mitochondria where it is thought to play important roles in mitochondrial proteostasis[26]. HTRA2 expression is upregulated by heat shock or activation of the p53 pathway[26]. Mice lacking HTRA2 or expressing inactive HTRA2 mutants display a neurodegenerative phenotype, suggesting that HTRA2 may be neuroprotective[26]. Further, inactivating mutations in HTRA2 have been implicated in PD[26,33]. Taken together, we hypothesized that HTRA proteins directly regulate the aggregation and clearance of a range of amyloid proteins.

Amyloid and other misfolded protein aggregates are highly resistant to degradation, and effective therapeutics that clear misfolded proteins are not available[15]. Agents that could reverse the formation of toxic α-syn species would be attractive disease-modifying therapies for PD and other synucleinopathies. Such agents could simultaneously reverse a toxic gain of function of the misfolded species and prevent further propagation of pathology via seeding, while also preserving the normal physiological function of α-syn. Here, we show that HTRA1, but not HTRA2, prevents and reverses fibrillization of α-syn, and that this treatment renders preformed α-syn seeds incapable of proteopathic seeding. This activity does not require HTRA1 to be proteolytically active. In contrast with previous studies[28], we find that α-syn remodeling does not require the HTRA1 PDZ domain, but can be entirely mediated via the proteolytically inactive HTRA1 protease domain. We find that HTRA1 can also prevent and degrade accumulations of TDP-43 and FUS, though activity against α-syn is more robust. We explored the mechanism of HTRA1-mediated disaggregation and found that HTRA1 confers remodeling by engaging the NAC core of the α-syn fibrils. Upon exposure of cells to preformed α-syn fibrillar seeds, elevated expression of proteolytically inactive HTRA1 prevents the triggering of α-syn aggregation, while knockdown of endogenous HTRA1 levels makes the cells more susceptible to α-syn seeding. Finally, treatment of α-syn with proteolytically inactive HTRA1 renders products that are non-toxic and incapable of seeding α-syn aggregation in primary mouse neurons. We demonstrate that HTRA1 can promote solubilization of a range of otherwise recalcitrant proteins, allowing for the clearance of the aggregates.

## Results

### HTRA1 can proteolyze α-synuclein, TDP-43, and FUS

HTRA1 has been demonstrated to disintegrate fibrillar tau, allowing for its subsequent clearance[28]. Further, HTRA1 has been shown to associate with microtubules and degrade tubulins, thereby inhibiting cell migration[28,34]. These findings have led to speculation that HTRA1 specifically regulates tau misfolding. We hypothesized that, because the amyloid fold is highly conserved[35], HTRA1 may be active against a range of amyloid and amyloid-like proteins beyond tau. Additionally, we sought to more broadly investigate HTRA1 activity against folded, intrinsically disordered, and fibrillar substrates. We were also curious to test the activity of HTRA2, which resides in the mitochondria. Mitochondrial dysfunction is an important aspect of Parkinson's disease (PD) pathophysiology and loss-of-function mutations in HTRA2 have been implicated in PD[26,33,36]. HTRA1 and HTRA2 are each comprised of an N-terminal domain, a protease domain, and a PDZ domain (Fig. 1a). The N-terminal domain of HTRA1 has a cleavable signal peptide (SP) followed by a fragment of insulin growth factor binding protein 7 (IGFBP), and a Kazal-type protease-inhibitor motif. While HTRA1 is primarily secreted into the extracellular space, ~20% of HTRA1 remains in the cytoplasm[27]. In contrast, the N-terminal domain of HTRA2 harbors a mitochondrial targeting sequence (MTS) followed by a transmembrane anchor that can be removed in processing[27].

We first sought to test HTRA1 and HTRA2 for their capacity to proteolyze and/or inhibit the aggregation of α-synuclein (α-syn), as well as TDP-43 and FUS. To directly assay activity, we employed recombinant proteins. Here, we used TDP-43-TEV-MBP-His$_6$ and GST-TEV-FUS constructs, where MBP and GST function as solubility tags (Fig. 1b)[37,38]. TDP-43 and FUS remain soluble for many hours with the tags appended, while aggregation proceeds rapidly upon cleavage of the MBP or GST solubility tags with TEV protease. α-Syn was purified as previously described[39].

We find that HTRA1 completely digests a 5-fold molar excess of monomeric α-syn within 24 h of incubation (Fig. 1c). We also note some autoproteolysis of HTRA1. In contrast, HTRA2 did not digest α-syn monomer. We next purified TDP-43-TEV-MBP and GST-TEV-FUS, which both form amyloid-like aggregates[37,38,40]. To initiate these reactions, we cleaved with TEV protease to liberate TDP-43 and FUS from the solubility tags, and then added HTRA1, HTRA2, or buffer (Fig. 1d, e). Both HTRA1 and HTRA2 displayed robust proteolysis of TDP-43. In contrast, while HTRA1 proteolyzed FUS, we observed somewhat weaker proteolysis of FUS by HTRA2. We noted no degradation of the free MBP or GST tags following TEV cleavage (Fig. 1d, e), or degradation of purified GST (Fig. S1A) by HTRA1 or HTRA2. In contrast, the intrinsically disordered casein protein was fully degraded by both HTRA1 and HTRA2 (Fig S1B), suggesting that HTRA proteins selectively proteolyze disordered proteins including casein, α-syn, TDP-43, and FUS, but not well-folded proteins including MBP and GST.

### HTRA1 inhibits TDP-43 and FUS aggregation

We were next curious if HTRA1 harbored chaperone activity toward these substrates in addition to its proteolytic activity. To detect aggregation of TDP-43 and FUS, we first employed turbidity assays (Fig. 1f–h). To assay inhibition independently of proteolysis, we included the proteolytically inactive variant HTRA1$^{S328A}$ (HTRA1SA). This variant harbors a mutation at the catalytic serine to alanine, which

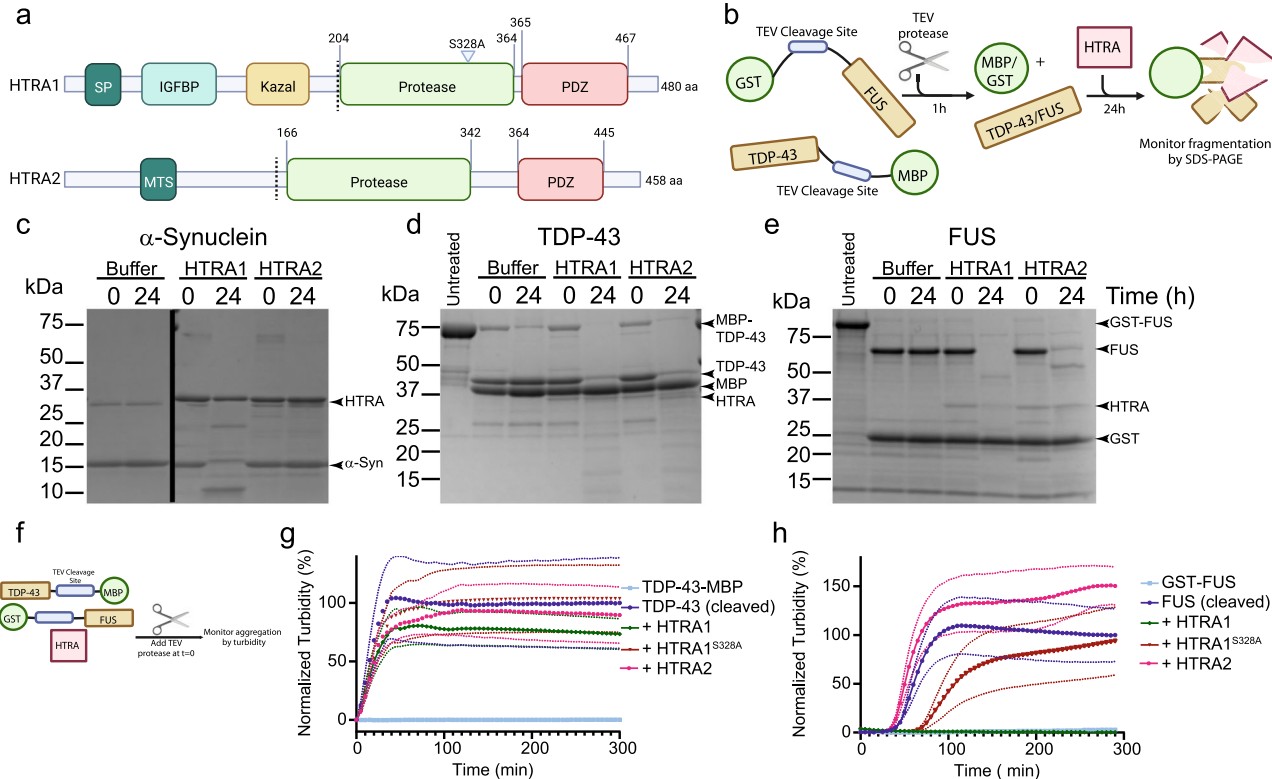

**Fig. 1 | HTRA1 can process diverse substrates for proteolysis. a** Domain architecture of HTRA1 and HTRA2. Signal peptide (SP), insulin-like growth factor binding protein (IGFBP), Kazal-like domain, and mitochondrial targeting sequence (MTS). Dashed lines indicate sites where ΔNTD constructs used in our studies begin. **b** Experimental setup for proteolysis experiments for TDP-43 and FUS. **c** α-synuclein monomer (25 µM) was treated with buffer, HTRA1, or HTRA2 (5 µM) for 24 h at 37 °C. Samples were then separated by SDS-PAGE and visualized with Coomassie blue staining ($N = 5$ independent experiments). **d** TDP-43-TEV-MBP (10 µM) was treated with TEV protease for 1 h at 37 °C, followed by treatment with buffer, HTRA1, or HTRA2 (2 µM) for 24 h at 37 °C. Samples were then processed by SDS-PAGE. Untreated lane is TDP-43-TEV-MBP with no added TEV or HTRA ($N = 5$ independent experiments). **e** GST-TEV-FUS (10 µM) was treated with TEV protease

for 1 h at 37 °C, followed by treatment with buffer, HTRA1, or HTRA2 (2 µM) for 24 h at 37 °C. Samples were then processed by SDS-PAGE. Untreated lane is GST-TEV-FUS with no added TEV or HTRA ($N = 5$ independent experiments). **f** Schematic of turbidity assay design. **g** TDP-43-TEV-MBP (10 µM) was incubated with buffer, HTRA1, HTRA1$^{S328A}$, or HTRA2 (10 µM). Reactions were initiated by addition of TEV protease at $t = 0$ and aggregation was monitored by turbidity ($N = 3$ independent experiments, means are shown as large symbols, SEM is shown as smaller symbols of the same color). **h** GST-TEV-FUS (10 µM) was incubated with buffer, HTRA1, HTRA1$^{S328A}$, or HTRA2 (10 µM). Reactions were initiated by addition of TEV protease and then aggregation was monitored by turbidity ($N = 3$ independent experiments, means are shown as large symbols, SEM is shown as smaller symbols of the same color).

ablates proteolysis and allows us to distinguish between proteolysis and remodeling by comparing the activities of HTRA1$^{WT}$ to HTRA1$^{S328A}$. We also included aldolase as a control for bulk protein effects. HTRA1 subtly delays TDP-43 aggregation when co-incubated at an equimolar ratio, though this effect is similar to that of aldolase (Fig. 1g and Supplementary Fig. 1C). HTRA1$^{S328A}$ and HTRA2 do not modulate TDP-43 aggregation under these conditions. We repeated these assays with a 3-fold and 5-fold molar excess of HTRA1 and observed a dose-dependent increase in inhibition of TDP-43 aggregation by HTRA1 and HTRA1$^{S328A}$ (Supplementary Fig. 1C). At a 5-fold molar excess of HTRA1, nearly complete inhibition of TDP-43 aggregation is achieved. HTRA1$^{S328A}$ shows a similar, though expectedly weaker, inhibitory effect. In contrast, even a 5-fold molar excess of HTRA2 has minimal effect on TDP-43 aggregation, despite its proteolytic activity against TDP-43 (Fig. 1d and Supplementary Fig. 1C).

We then tested inhibition of FUS aggregation and found that HTRA1 activity against FUS is more potent than against TDP-43. Here, an equimolar ratio of HTRA1 achieved complete inhibition of FUS aggregation, while HTRA2 appeared to accelerate aggregation despite its proteolytic activity against FUS (Fig. 1e, h). Even HTRA1$^{S328A}$, which lacks proteolytic activity, considerably slowed aggregation (Fig. 1h), with a 5-fold molar excess completely inhibiting FUS aggregation (Supplementary Fig. 1D). To further validate these effects, we performed sedimentation assays where solubility was assessed in the

presence or absence of HTRA. Here, a 20-fold molar excess of HTRA1, HTRA1$^{S328A}$, HTRA2, or a buffer control was mixed with TDP-43-TEV-MBP while a 5-fold molar excess of HTRA was incubated with GST-TEV-FUS (Supplementary Fig. 1E–H). After 24 h of incubation, whereas ~80% of TDP-43 or nearly 100% of FUS ordinarily partition to the insoluble fraction, HTRA1 co-incubation prevents any detectable accumulation of insoluble TDP-43 or FUS, presumably in part due to proteolysis. However, HTRA1$^{S328A}$ also strongly preserved solubility, with ~60% of TDP-43 and ~90% of FUS remaining soluble. We therefore conclude that both HTRA1 and HTRA2 can degrade TDP-43 and FUS, while only HTRA1 can process α-syn for proteolysis. Further, as indicated by the similar inhibitory activities of HTRA1$^{WT}$ and HTRA1$^{S328A}$, we can conclude that inhibition of TDP-43 and FUS aggregation can occur in a proteolytically-independent fashion, with inhibition of FUS aggregation being more potent. Interestingly, given the differences in proteolysis and inhibition we observed, the HTRA proteins appear to operate via two distinct mechanisms. To further explore these features, we focused on α-syn because we found that activity against α-syn aggregation was the most potent.

## HTRA1 prevents α-synuclein amyloidogenesis and preserves α-synuclein solubility

We next sought to elucidate the activity of HTRA1 and HTRA2 in antagonizing α-syn misfolding. To monitor amyloid formation, we

used the amyloid-binding dye ThioflavinT (ThT) to detect α-syn amyloidogenesis. α-Syn monomer assembles into fibrils rapidly, with fibrillization complete after ~48 h of agitation. When α-syn fibrillization is conducted in the presence of HTRA1, with a 5-fold molar excess of α-syn monomer, the ThT signal is decreased by ~80% as compared to aggregation achieved in the absence of HTRA1 (Fig. 2a). Addition of the proteolytically inactive HTRA1[S328A] variant achieved a similar level of inhibition, indicating that inhibition of amyloidogenesis by HTRA1[S328A] can proceed in a protease-independent fashion, and is not simply due to degradation of α-syn monomer. HTRA2 activity is notably weaker, achieving only ~30% inhibition of α-syn amyloidogenesis. Similar effects were achieved after a 72 h incubation (Supplementary Fig. 2A). To assess if HTRA1 is active against mutations associated with familial PD, we performed similar experiments using α-syn[A53T][41]. We observed similar trends, with strong inhibition of α-syn[A53T] aggregation by HTRA1 and HTRA1[S328A], and minimal inhibition by HTRA2 (Fig. 2b). Our results suggest that both wild-type and α-syn[A53T] can be substrates for HTRA1, but that HTRA2 cannot inhibit aggregation of α-syn or α-syn[A53T].

We hypothesized that the decreased formation of ThT-reactive species was due to preserved solubility of the α-syn monomer. To probe this, we first monitored solubility using sedimentation assays where solubility was assessed in the presence or absence of HTRA. Here, a 5-fold molar excess of α-syn monomer was incubated with HTRA1, HTRA1[S328A], HTRA2, aldolase, or a buffer control (Fig. 2c, d). After 24 h of incubation, whereas 20% of α-syn ordinarily partitions to the insoluble fraction, HTRA1 co-incubation prevents any detectable accumulation of insoluble α-syn (Fig. 2d and Supplementary Fig. 2B, C). This protective effect persisted for 72 h of incubation, with less than 10% of α-syn accumulating in the insoluble fraction, by which time nearly 60% of α-syn was found in the pellet in the absence of HTRA1 (Fig. 2d). However, we did note proteolysis of both HTRA1[WT] and α-syn (Fig. 2c). Because we cannot rule out that this inhibitory activity of HTRA1[WT] could alternatively be explained by degradation of α-syn monomer, we again assessed solubilization independently of proteolysis by also testing HTRA1[S328A]. When incubated with HTRA1[S328A], we find that α-syn is retained in the soluble fraction even when there is no proteolytic cleavage, with ~80% of α-syn remaining soluble after 72 h (Fig. 2c, d). Addition of either HTRA2 or aldolase has no inhibitory effect on α-syn aggregation, although HTRA2 undergoes complete auto-proteolysis within the 72 h incubation (Fig. 2c). We were surprised to note some accumulation of HTRA1[S328A] in the insoluble fraction, so we also explored HTRA solubility in the absence of substrate (Fig. 2e). We find that both HTRA1 and HTRA1[S328A] largely partition to the insoluble fraction when incubated without substrate (Fig. 2e), while nearly all HTRA1 and HTRA1[S328A] are found in the soluble fraction in the presence of α-syn (Fig. 2c).

Using negative stain transmission electron microscopy, we confirmed that treatment of α-syn monomer with HTRA1 prevents the formation of α-syn fibrils, with just a small amount of amorphous material accumulating (Fig. 2f and Supplementary Fig. 2D). HTRA1[S328A] also modulates α-syn fibrillization, whereby the treated products are less abundant and appear more diffuse as compared to the tightly-packed appearance of the untreated fibrils, correlating with the ThT results which indicate decreased amyloid content of the treated products (Fig. 2a). Treatment with HTRA2 did not result in any apparent changes to α-syn fibril morphology. Thus we conclude that HTRA1, but not HTRA2, can prevent α-syn[WT] and α-syn[A53T] from forming amyloid fibrils, and that α-syn and HTRA1 form a stable complex that preserves the solubility of both proteins. Though proteolysis may be responsible for the inhibitory activity noted for proteolytically active HTRA1[WT], through the use of the proteolytically inactive HTRA1[S328A] variant, we can confirm that this activity is proteolytically independent and mediated via a direct interaction between the two proteins.

## HTRA1 treatment renders α-synuclein seeding incompetent

When preformed fibrillar (PFF) α-syn is added exogenously to mammalian cell cultures or via intrastriatal inoculation of mice, these PFFs enter the cell and initiate the seeding and aggregation of endogenous soluble α-syn[4,5,42]. To monitor this process, we used a HEK293T FRET biosensor cell line engineered to stably co-express cyan fluorescent protein (CFP)-tagged α-syn and yellow fluorescent protein (YFP)-tagged α-syn[43]. Upon addition of α-syn PFFs to the cell culture medium, PFFs trigger aggregation of the fluorescent α-syn, which can be observed by fluorescence microscopy and measured by FRET[43]. Based on our findings that co-incubation of α-syn monomer with HTRA1 prevents α-syn fibrillization and preserves α-syn solubility, we hypothesized that HTRA1 treatment would also render α-syn incapable of forming species that seed α-syn aggregation in HEK293T biosensor cells. To test this idea, we agitated α-syn monomer in the presence of HTRA1, HTRA1[S328A], aldolase, or buffer with a 5-fold molar excess of α-syn monomer. We then applied the reaction products to HEK293T α-syn FRET biosensor cells. Cells were analyzed by fluorescence microscopy or flow cytometry to assess α-syn aggregation (Fig. 3). We first sought to confirm that the fibrils we generated could be internalized into cells and trigger seeding. To do so, we formed PFFs from Alexa-568 labeled α-syn monomer. We also agitated labeled α-syn monomer in the presence of HTRA1 or HTRA1[S328A]. We then assessed internalization by flow cytometry and find that Alexa-568-α-syn was internalized in nearly 40% of cells. Co-incubation with HTRA1 or HTRA1[S328A] did not significantly modulate α-syn uptake (Fig. 3b and Supplementary Fig. 3). For further experiments, we employed unlabeled α-syn to avoid confounding effects due to the fluorophore. Application of 50 nM PFFs was sufficient to induce robust seeding of the biosensor cells, with abundant puncta throughout the cell population and a strong FRET signal as detected by flow cytometry (Fig. 3c–e). However, pre-treatment with HTRA1 or HTRA1[S328A] nearly completely abolished α-syn seeding capacity (Fig. 3c–e). Application of higher concentrations of 100, 200, and 400 nM treated α-syn gave similar results (Supplementary Fig. 3D). Quantification of these effects by flow cytometry indicates that HTRA1 or HTRA1[S328A] treatment prevents formation of any seeding-competent PFFs (Fig. 3c and Supplementary Fig. 3D, E). Similar inhibition of seeding was observed with Alexa-568 labeled α-syn (Supplementary Fig. 3A, C). Treatment with HTRA1[S328A] also markedly reduced the seeding competence of the α-syn. In contrast, aldolase treatment had no significant effect (Fig. 3d and Supplementary Fig. 3F). Similar results were achieved when treating α-syn[A53T] with HTRA1 or HTRA1[S328A] (Fig. 3e and Supplementary Fig. 3G). Thus, HTRA1 robustly inhibits the conversion of α-syn and α-syn[A53T] monomer to a seeding competent form, and this activity can be achieved not just through proteolytic degradation of α-syn, but also in a proteolytically-independent manner with HTRA1[S328A]. Furthermore, the products of HTRA1 remodeling and proteolysis cannot serve as seeds to nucleate and propagate α-syn aggregation.

## The protease domain of HTRA1 is necessary and sufficient for substrate remodeling

HTRA proteases share many features with classical serine proteases including trypsin and chymotrypsin[27]. However, HTRA proteases are unique because their activity is finely tuned and can be reversibly switched on and off, unlike classical serine proteases. This distinct structural and functional plasticity is thought to be mediated by the PDZ domain of HTRA, and it is thought that HTRA activity is regulated by the binding of peptides to the PDZ domain[27,29].

To explore how this mechanism ultimately dictates HTRA activity against α-syn, we employed a series of constructs with the protease or PDZ domain deleted (Fig. 4a). To probe a possible direct interaction between HTRA and α-syn, as suggested by our sedimentation assay results (Fig. 2), we allowed fibrillization to proceed in the presence of HTRA1, HTRA1[S328A], or HTRA2 and monitored complex formation using

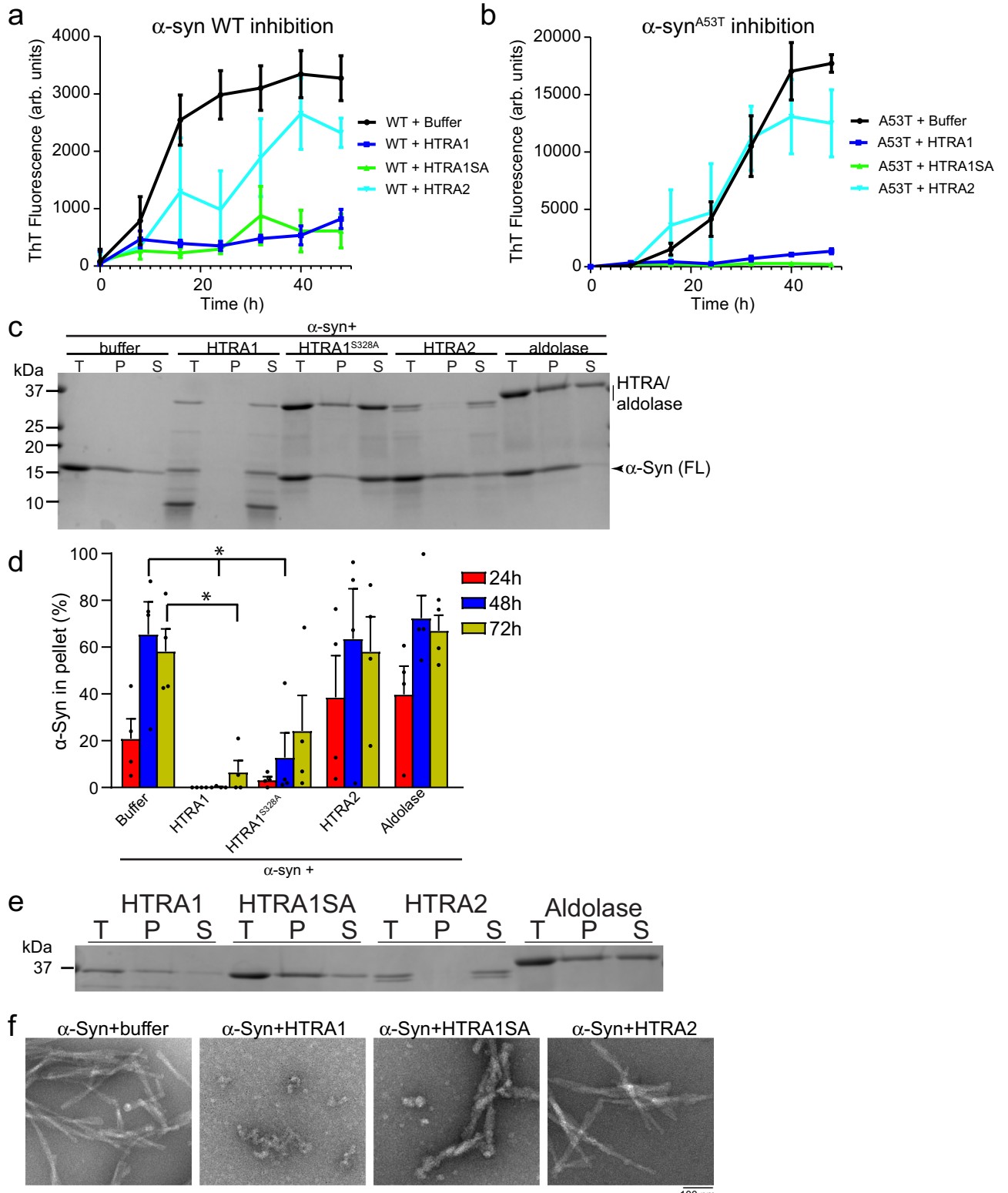

**Fig. 2 | HTRA1 prevents α-Syn amyloidogenesis and preserves α-Syn solubility.**
**a** α-Syn monomer (25 μM) was incubated with buffer, HTRA1, HTRA1^S328A, HTRA2, or aldolase (5 μM) for 48 h. Amyloid content was assessed by ThioflavinT (ThT) fluorescence every 8 h (N = 3 independent experiments, data represent means ± SEM. **b** Experiments were performed as in **a** but with α-syn^A53T (N = 3 independent experiments, data represent means ± SEM. **c** α-Syn (25 μM) monomer was incubated with HTRA1, HTRA1^S328A, HTRA2, or aldolase (5 μM) at 37 °C. At the indicated time points, samples were removed and fibrillization was assessed by sedimentation. (T total, P pellet, S soluble). SDS-PAGE gel for 72 h time point is shown (N = 4

independent experiments). **d** Quantification of sedimentation analysis from **c**. Values at 24, 48, or 72 h were compared to buffer control at the same time point using a one-way ANOVA with a Dunnett's multiple comparisons test (N = 4 independent experiments, bars are means ± SEM, *p = 0.0195 (buffer vs. HTRA1, 72 h), *p = 0.0368 (buffer vs. HTRA1^S328A, 48 h). **e** Experiments were performed as in **c**, but in the absence of α-syn to monitor solubility of HTRA independently. SDS-PAGE gel for 24 h time point is shown (N = 4 independent experiments). **f** Fibrillization reactions performed as in **a**–**c** and processed for EM. Representative images are shown. Scale bar, 100 nm.

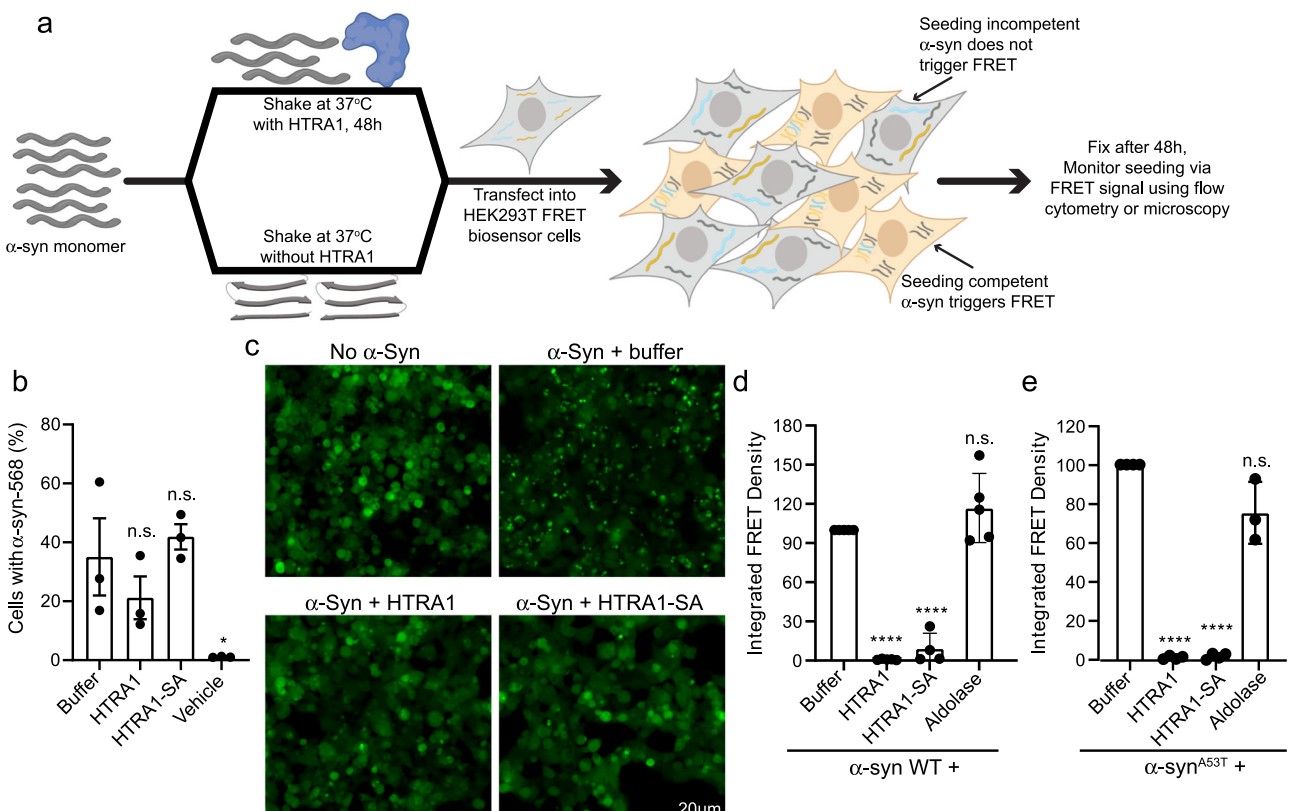

**Fig. 3 | HTRA1 prevents α-Syn from forming seeding competent species.**
**a** Schematic showing experimental design. **b** α-Syn-Alexa568 (25 μM) monomer was incubated with buffer, HTRA1, or HTRA1$^{S328A}$ (5 μM) at 37 °C for 48 h. Reaction products were transfected into HEK293T biosensor cells (50 nM α-syn in media). 48 h following treatment, cells were analyzed by flow cytometry for quantification of internalization based on Alexa-568 signal. Values were compared to control reactions with α-syn alone using a one-way ANOVA with a Dunnett's multiple comparisons test ($N = 3$ independent experiments, data represent means ± SEM, *$p = 0.037$). **c** α-Syn (25 μM) monomer was incubated with buffer, HTRA1, or HTRA1$^{S328A}$ (5 μM) at 37 °C for 48 h. Reaction products were transduced into

HEK293T biosensor cells (50 nM α-syn in media). Cells were assessed by microscopy 48 h following treatment. **d** Cells from **c** were analyzed by flow cytometry and integrated FRET density was calculated. Values were compared to control reactions with α-syn alone using a one-way ANOVA with a Dunnett's multiple comparisons test ($N = 5$ for buffer, HTRA1, and aldolase; $N = 4$ for HTRA1-SA, biological replicates are shown as dots, bars represent means ± SEM, ****$p < 0.0001$). **e** Experiments were performed as in **c** but with α-syn$^{A53T}$. Values were compared to control reactions with α-syn alone using a one-way ANOVA with a Dunnett's multiple comparisons test. Data represent means ± SEM, ($N = 4$ (buffer) and 3 (HTRA1, HTRA1-SA, and aldolase) independent experiments, ****$p < 0.0001$)).

native PAGE (Supplementary Fig. 4A). In the presence of HTRA1$^{S328A}$, we note a distinct smear at a higher molecular weight than that of HTRA1$^{S328A}$ or α-syn monomer alone, corresponding to likely complex formation between HTRA1$^{S328A}$ and α-syn. We observe a sharp band for HTRA2 at its expected molecular weight, indicating no complex formation with α-syn, though there does appear to be some protein trapped in the wells of the gel, suggestive of formation of some higher order complexes. In the presence of α-syn, HTRA1$^{WT}$ also forms a smear, but at an intermediate molecular weight, and of decreased intensity. We excised this band from the gel and confirmed the presence of both HTRA1 and α-syn by mass spectrometry (Supplementary Fig. 4B). This suggests that HTRA1$^{WT}$ or HTRA1$^{S328A}$, but not HTRA2, form a stable complex with α-syn. Further, this interaction with HTRA1 appears to partially protect α-syn from proteolysis.

Using a fluorescein isothiocyanate (FITC)-casein model substrate, we next tested if the PDZ domain was required for proteolysis (Fig. 4b). Here, FITC fluorescence is quenched due to fusion to casein and upon degradation of casein, this self-quenching is diminished and FITC fluorescence increases. We observe that both HTRA1 and the protease domain alone (HTRA1ProD) robustly digest the FITC-casein substrate, though digestion is more efficient with full-length HTRA1. We confirmed these results also using α-syn monomer as a substrate (Fig. 4c). We also compared the proteolytic efficiency of HTRA1 and HTRA1ProD, and find that the time course of proteolytic

degradation is fairly similar for the two constructs, suggesting that deletion of the PDZ domain does not substantially impact proteolysis (Supplementary Fig. 4C). We hypothesized that due to the prominent role of PDZ domains in mediating protein-protein interactions[32], the PDZ domain would be essential for binding and suppressing amyloidogenesis. HTRA1ProD can completely inhibit the formation of ThT-reactive species (Fig. 4d) and preserve α-syn monomer solubility similarly to HTRA1 (Fig. 4e and Supplementary Fig. 4D). Interestingly, these effects also do not require HTRA1 proteolytic activity, as HTRA1ProD$^{S328A}$ shows similar levels of activity. Further, the isolated PDZ construct (HTRA1PDZ) had no effect on amyloidogenesis and only weakly preserved α-syn solubility (Fig. 4d, e). To further corroborate these results, we applied these remodeled products to the FRET biosensor cells and found that again, HTRA1ProD restricts the formation of seeding-competent species in a proteolysis-independent fashion while HTRA1PDZ only weakly inhibits seeding (Fig. 4f and Supplementary Fig. 4E). Finally, to investigate if this activity is mediated by a direct interaction, we monitored binding with pull-down assays (Fig. 4g, h). Supporting our earlier results, we again note strong binding by HTRA1$^{S328A}$ and HTRA1ProD$^{S328A}$, while HTRA1PDZ only weakly binds monomeric α-syn. Further, although HTRA2 has only limited remodeling activity against α-syn, it binds monomeric α-syn with similar affinity to HTRA1$^{S328A}$, indicating that chaperone activity observed by the HTRA proteins is not merely due

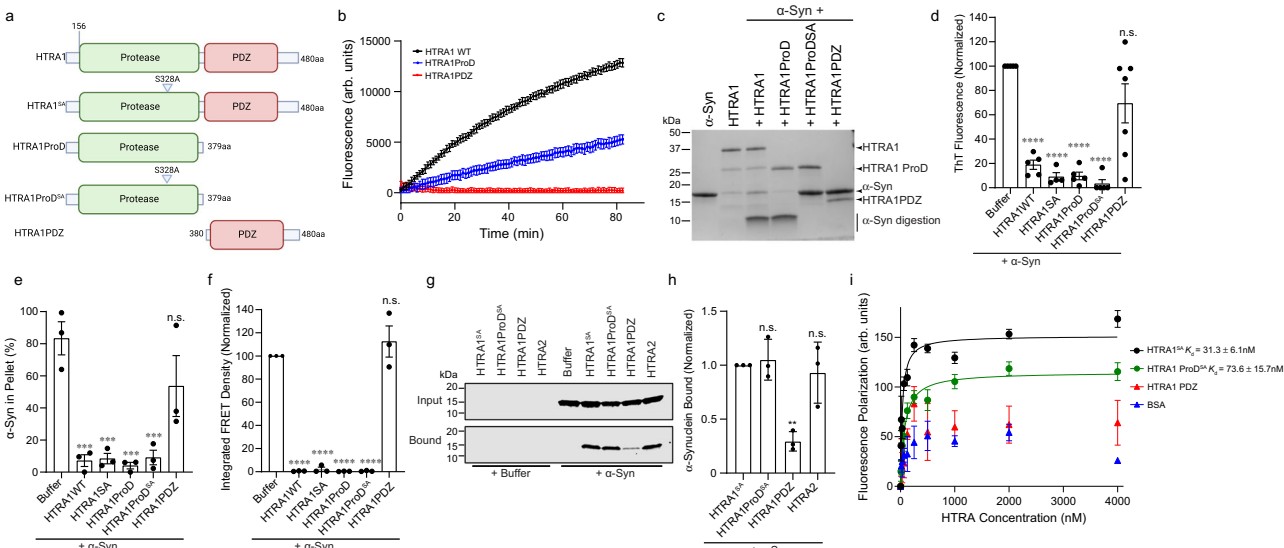

**Fig. 4 | The protease domain of HTRA1 is necessary and sufficient for remodeling α-syn. a** Constructs used in these experiments. **b** HTRA1 (Black), HTRA1-ProD (blue), or HTRA1PDZ (red) (2 μM) were incubated with FITC-casein (10 μM). FITC-casein degradation was monitored by fluorescence. $N = 3$ independent experiments, representative data from one trial is shown. Error bars show SEM for this technical triplicate. **c** α-Syn (25 μM) monomer was treated with buffer, HTRA1, HTRA1ProD, HTRA1ProDSA, or HTRA1PDZ (5 μM) for 24 h at 37 °C. Samples were processed by SDS-PAGE to assess α-syn proteolysis ($N = 3$ independent experiments). **d** α-Syn monomer (25 μM) was incubated with buffer or indicated HTRA1 construct (5 μM) for 48 h. Amyloid content was assessed by ThioflavinT (ThT) fluorescence. $N = 5$ for buffer, HTRA1WT, HTRA1ProD, HTRA1ProDSA, and HTRA1PDZ; $N = 4$ for HTRA1SA, and $N = 7$ for HTRA1PDZ, biological replicates are shown as dots, bars are means ± SEM (****$p < 0.0001$ by one-way ANOVA with Dunnett's multiple comparisons test compared to buffer). **e** Sedimentation assays with α-syn monomer (25 μM) and the indicated HTRA1 construct (5 μM) at 37 °C.

After 48 h, samples were assessed by sedimentation and quantified by SDS-PAGE. $N = 3$ independent experiments, bars represent means ± SEM, ***$p = 0.0004$ (buffer vs HTRA1WT), 0.0004 (buffer vs HTRA1SA), 0.0003 (buffer vs HTRA1ProD), 0.0005 (buffer vs HTRA1ProDSA) by one-way ANOVA with Dunnett's multiple comparisons test. **f** Samples from **d** were transduced into HEK293T biosensor cells. FRET was assessed by flow cytometry after 48 h ($N = 3$ independent experiments, bars represent means ± SEM, ****$p < 0.0001$ by one-way ANOVA with Dunnett's multiple comparisons test as compared to α-syn alone). **g** His-tagged HTRA constructs (10 μM) were immobilized on Ni-NTA resin and incubated with α-syn monomer (20 μM) overnight. Input and bound fractions were processed by immunoblotting ($N = 3$ independent experiments). **h** Experiments from **g** were quantified and normalized to the HTRA1$^{S328A}$ condition. $N = 3$, bars represent means ± SEM, **$p = 0.0052$, n.s. $p > 0.05$ by one-way ANOVA with Dunnett's multiple comparisons test, as compared to HTRA1$^{SA}$). **i** Fluorescence polarization binding analysis ($N = 3$ independent experiments, means are shown as dots, bars show ± SEM).

to the effects of binding. These results are in contrast to those observed in the native PAGE assays (Supplementary Fig. 4A), where no complex formation is observed for HTRA2. This is possibly due to the differing timescales of the two experiments, and suggests that while HTRA2 can bind α-syn monomer, binding alone is insufficient to prevent aggregation, and aggregation can still occur on a longer timescale despite complex formation. To further investigate and quantify these interactions, we performed fluorescence polarization assays (Fig. 4i). We found that HTRA1$^{S328A}$ binds monomeric α-syn with a $K_d$ of ~30 nM while HTRA1ProD$^{S328A}$ binds α-syn monomer with a $K_d$ of ~74 nM. No binding was detected to a BSA control or HTRA1PDZ, though it is important to note that due to the small size of HTRA1PDZ, a binding signal in our assay may be too low for detection for this protein. In sum, our results suggest that binding to α-syn monomer is insufficient to inhibit aggregation, but that HTRA1 is instead conferring a distinct remodeling activity. We can conclude that HTRA1 chaperoning of α-syn relies on direct interaction between the protease domain and α-syn, and that the protease domain is necessary and sufficient for this interaction. Further, this activity does not depend upon the proteolytic activity of this domain.

We performed similar experiments to determine if these same features apply to inhibition of TDP-43 and FUS aggregation (Supplementary Fig. 5). We find that HTRA1ProD can robustly proteolyze both TDP-43 and FUS, and that HTRA1ProD$^{S328A}$ preserves the solubility of both TDP-43 and FUS, though higher concentrations of HTRA are required for solubilization of TDP-43 and FUS as compared to α-syn. We also find that, like with α-syn, the PDZ domain alone cannot inhibit TDP-43 or FUS aggregation. Therefore, we conclude that HTRA1 can

proteolyze and inhibit the aggregation of a range of substrates, though activity is more robust against amyloid substrates such as α-syn as compared to substrates such as TDP-43 and FUS, which have disordered regions as well as globular regions.

## HTRA1 dissolves *preformed* α-synuclein fibrils and renders them seeding incompetent

We next aimed to assess if HTRA1 could not just prevent α-syn from forming seeding competent species, but also dissolve α-syn PFFs and diminish their seeding capacity. Here, we treated mature PFFs with HTRA proteins and monitored the biophysical properties of the treated PFFs as well as their seeding capacity (Fig. 5a). Treatment of α-syn PFFs with HTRA decreased the ThT signal by ~60% for HTRA1 and 40% for HTRA1$^{S328A}$ (Fig. 5b). Disaggregation was also assessed by sedimentation assay (Fig. 5c, d). Here, following the treatment of PFFs with HTRA1, the reaction products were partitioned to a soluble and insoluble fraction. Upon treatment with HTRA1, we noted a decrease in total α-syn, presumably due to proteolysis. However, this decrease in α-syn was primarily in the soluble fraction and not the pellet fraction. In contrast, upon treatment with HTRA1$^{S328A}$, we note a decrease primarily in the insoluble pellet fraction. This suggests that HTRA1 disaggregase activity is preferential for the insoluble species, while soluble species are favored for proteolysis. Remodeling was also noted by electron microscopy. HTRA1 treatment led to the fibrils adopting a more diffuse appearance while HTRA1$^{S328A}$ treatment yielded amorphous accumulations that did not resemble fibrils (Fig. 5e and Supplementary Fig. 6A). HTRA1 or HTRA1$^{S328A}$ treatment reduced seeding capacity of these products by ~30–40% when applied to FRET

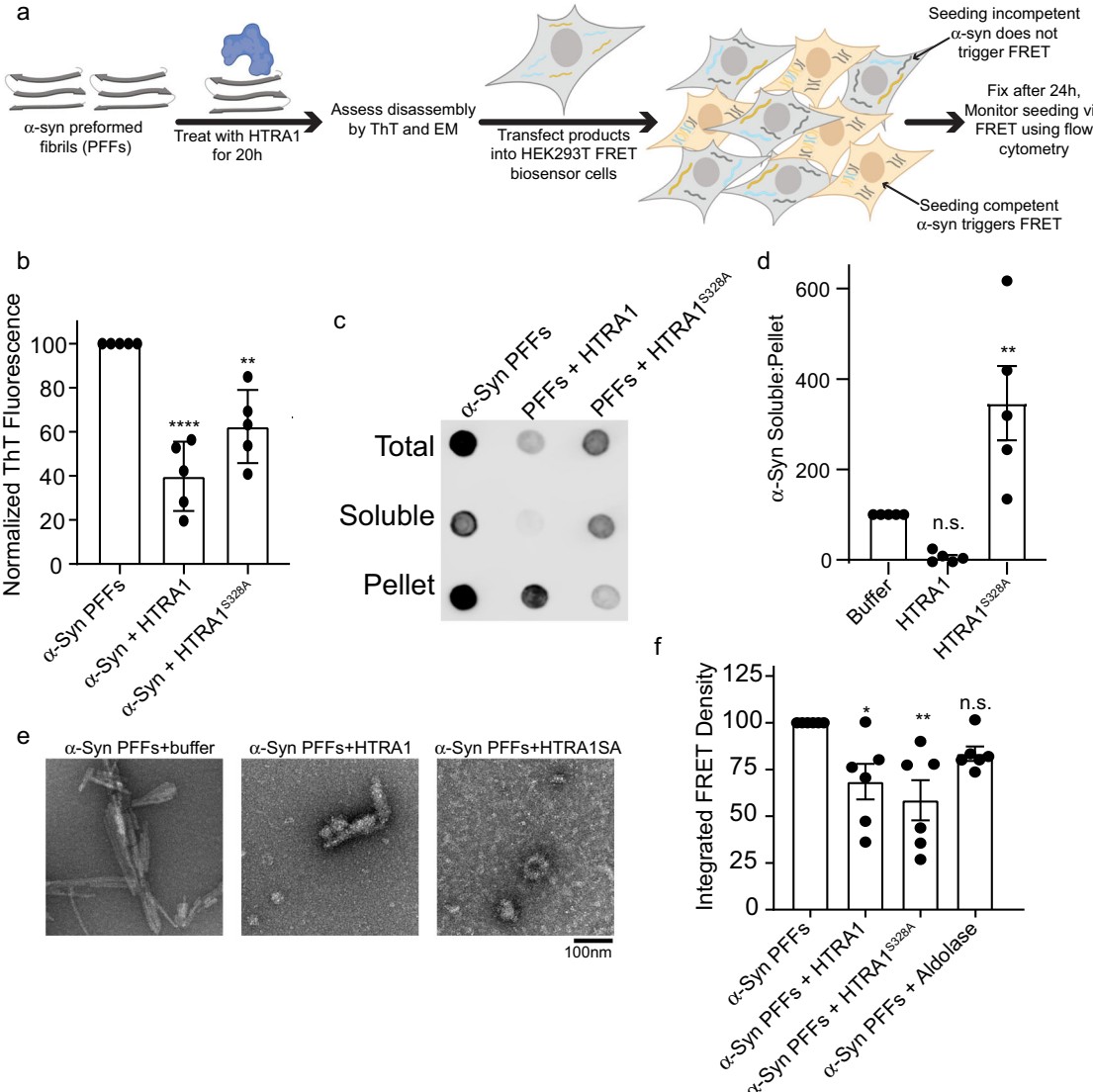

**Fig. 5 | HTRA1 treatment remodels α-Syn PFFs to a seeding incompetent form.**
**a** Schematic showing experimental design. **b** α-Syn PFFs (10 μM) were treated with buffer, HTRA1, or HTRA1S328A (100 μM) for 24 h and amyloid content was assessed by ThioflavinT (ThT) fluorescence. Values are compared to buffer treatment using a one-way ANOVA with a Dunnett's multiple comparisons test ($N = 5$, biological replicates are shown as dots, bars are means ± SEM, **$p = 0.00135$, ****$p < 0.0001$). **c** α-Syn PFFs (5 μM) were treated with buffer, HTRA1, or HTRA1S328A (100 μM) for 48 h and then partitioned into total, soluble, and insoluble fractions. Samples were then spotted onto a nitrocellulose membrane and α-syn was visualized by immunoblotting. Representative image is shown. **d** Quantitation of experiments shown in

**c.** Values are normalized to α-syn alone condition and compared to this treatment using a one-way ANOVA with a Dunnett's multiple comparisons test ($N = 5$, biological replicates are shown as dots, bars are means ± SEM, **$p = 0.0059$). **e** Reactions from **b** were processed for EM. Scale bar, 100 nm. **f** α-Syn PFFs were treated as in **b** and the remodeled products were transduced into HEK293T biosensor cells (10 nM PFFs) using lipofectamine. 24 h after transduction, cells were harvested and integrated FRET density was measured by flow cytometry. Values are compared to buffer treatment using a one-way ANOVA with a Dunnett's multiple comparisons test ($N = 6$, biological replicates are shown as dots, bars represent means ± SEM, *$p = 0.0188$, **$p = 0.0022$).

biosensor cells (Fig. 5f and Supplementary Fig. 6B). To further confirm that disaggregation is independent of proteolysis, and the effects we observe are not simply due to proteolysis of fibrillar α-syn, we performed a proteolysis experiment to compare HTRA1WT activity against the monomeric and fibrillar forms of α-syn. Here, we observe nearly complete proteolysis of monomeric α-syn, with minimal proteolysis of the fibrillar form (Supplementary Fig. 6C). Therefore, we can conclude that proteolysis of α-syn fibrils is not the primary contributor to the observed disaggregation of α-syn PFFs, and instead these activities rely upon the disaggregase activity of HTRA1. In sum, we conclude that HTRA1 can remodel α-syn PFFs, decreasing their amyloid content, disrupting their morphology, and decreasing their seeding capacity. Further, through the use of HTRA1S328A, we demonstrate that this remodeling activity does not require proteolytic activity.

## HTRA1 disaggregates α-synuclein fibrils by specifically targeting the NAC domain

To better understand the mechanism by which HTRA1 remodels α-syn fibrils at higher resolution, we performed proteolysis experiments followed by identification of the cleavage products by liquid chromatography/mass spectrometry (LC/MS). First, we incubated α-syn monomer with HTRA1 and analyzed the cleavage pattern by LC/MS (Fig. 6a). We found that cleavage occurs throughout the α-syn sequence, with cleavage enriched in the nonamyloid component (NAC) domain, residues 61-95, a domain known to play a critical role in catalyzing α-syn oligomerization and fibrillization[44]. The cleavage sites we identified are consistent with previously reported trends in HTRA1 proteolytic cleavage, where HTRA1 preferentially cleaves following residues such as valine and threonine[45]. Next, to better understand

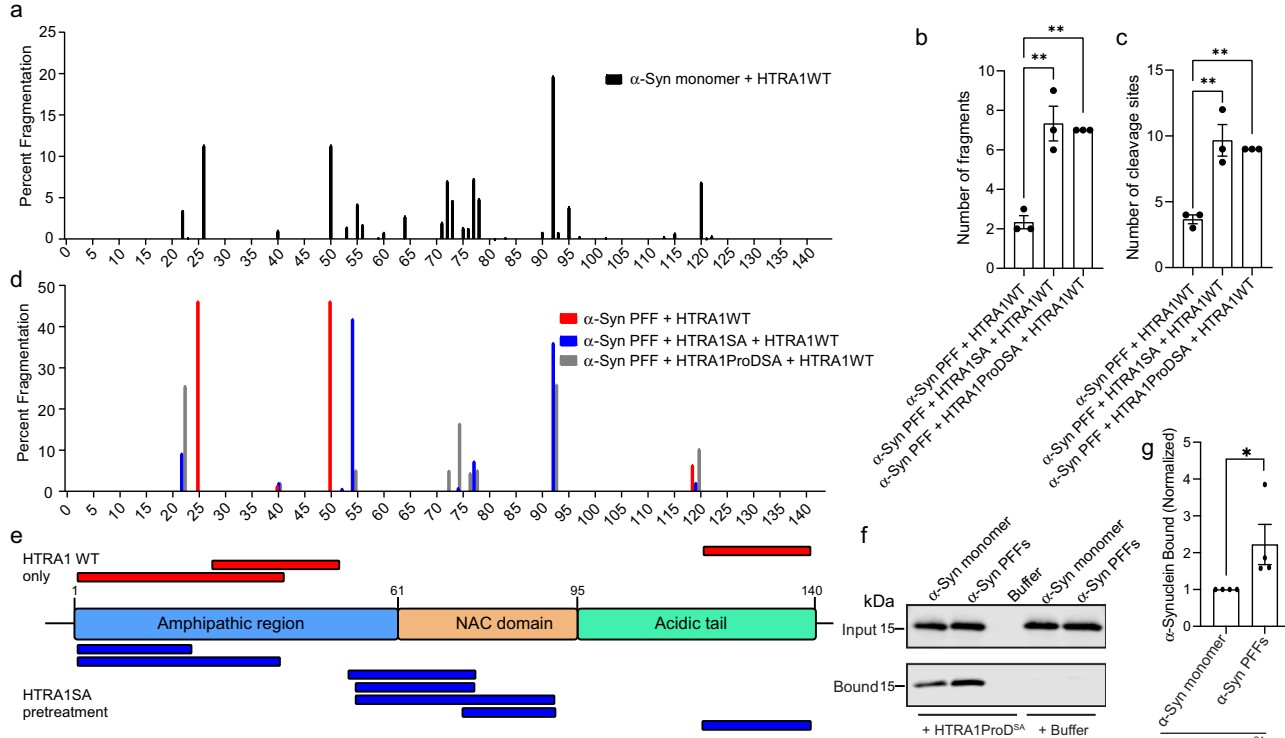

**Fig. 6 | HTRA1 disaggregation promotes solubilization and proteolysis of the NAC domain. a** α-Syn monomer (25 μM) was treated with HTRA1 (5 μM) for 3 h. Samples were then analyzed by LC/MS. **b** α-Syn PFFs (5 μM) were pre-treated with buffer, HTRA1$^{S328A}$, or HTRA1ProD$^{S328A}$ (50 μM) for 2 h, followed by addition of HTRA1 (2.5 μM) for 3 h. Samples were then analyzed by LC/MS. The total number of fragments detected or (**c**) total number of cleavage sites are shown. Values were compared to α-syn PFF + HTRA1WT using a one-way ANOVA with Dunnett's multiple comparisons test ($N = 3$, biological replicates are shown as dots, bars represent means ± SEM, **$p = 0.0019$ (HTRA1SA, panel **b**), **$p = 0.0035$ (HTRA1ProDSA, panel **b**), **$p = 0.0011$ (HTRA1SA, panel **c**) and **$p = 0.0016$ (HTRA1ProDSA, panel **c**). **d** Quantification of the relative abundance of fragmentation at specific cleavage sites from experiments shown in **b**, **c**. Percent fragmentation at specific sites was normalized to the sum of the peak areas for each condition ($N = 3$, representative data from one replicate is shown). **e** Map of α-syn PFF proteolysis. Fragments identified upon cleavage with HTRA1 alone (red) and after adding a pre-treatment step with HTRA1$^{S328A}$ prior to proteolysis (blue). **f** His-HTRA1ProD$^{SA}$ (10 μM) was immobilized on Ni-NTA resin and incubated with α-syn monomer or pre-formed fibrils (PFFs) (20 μM) overnight. Beads were then washed and the input and bound fractions were processed by immunoblotting using an α-syn antibody. **g** Experiments from **f** were quantified and normalized to the α-syn monomer condition. Values are compared to control reactions with α-syn monomer using a one-way ANOVA with a Dunnett's multiple comparisons test ($N = 4$, biological replicates are shown as dots, bars represent means ± SEM, *$p = 0.0215$).

how conversion to the amyloid form modulates HTRA1 activity, we performed similar experiments with α-syn PFFs. Here, to more clearly identify the key cleavage sites, we pre-treated the PFFs with HTRA1$^{S328A}$ or HTRA1ProD$^{S328A}$ to render the PFFs more susceptible to proteolysis by active HTRA1 and then analyzed the fragments by LC/MS. We first analyzed the number of fragments produced and found that pre-treatment with HTRA1$^{S328A}$ or HTRA1ProD$^{S328A}$ rendered the PFFs more susceptible to fragmentation than treatment with HTRA1 alone (Fig. 6b, c). Analysis of the fragmentation pattern indicated that pre-treatment with HTRA1$^{S328A}$ or HTRA1ProD$^{S328A}$ resulted in more cleavage sites and greater overall fragmentation than treatment with HTRA1 alone (Fig. 6d, e and Supplementary Fig. 7). Further, pre-treatment with HTRA1PDZ resulted in cleavage patterns similar to those from treatment with HTRA1$^{WT}$ alone, with very few cleavage sites in the NAC domain (Supplementary Fig. 7). Analysis of the specific cleavage products indicated that treatment of fibrils with HTRA1$^{WT}$ alone resulted in cleavage at three primary positions in the α-syn sequence, all outside the NAC domain that is otherwise susceptible to cleavage when α-syn is in the monomeric form. In contrast, the addition of an HTRA1$^{S328A}$ pre-treatment step resulted in several new cleavage sites within the NAC domain (Fig. 6d, e). When PFFs are not pre-treated with HTRA1$^{S328A}$, this region remains resistant to cleavage. To further confirm the interaction between HTRA1 and α-syn, and to investigate if disaggregation can be attributed to binding of the monomeric form of α-syn, we compared the binding affinity of HTRA1 to monomeric and fibrillar α-syn. Using a pull-down assay, we observe that binding of HTRA1ProD$^{S328A}$ to α-syn fibrils is significantly tighter than binding to monomeric α-syn (Fig. 6f, g). This suggests that HTRA1 preferentially binds α-syn fibrils and cleaves α-syn monomer in the NAC domain, and treatment with HTRA1$^{S328A}$ or HTRA1ProD$^{S328A}$ mediates disaggregation by making the NAC domain protease-accessible. However, upon amyloidogenesis, this region becomes protected and resistant to HTRA1 cleavage. To enable cleavage even in the fibrillar state, the protease domain of HTRA1 directly engages α-syn fibrils to mediate disaggregation, thereby allowing proteolytic cleavage to proceed (Fig. 7).

## Overexpression of HTRA1 prevents α-syn PFFs from seeding aggregation of endogenous α-syn

We next sought to determine if HTRA1 expression could protect against α-syn seeding, or if HTRA1 pretreatment of the PFFs was required. We transfected HEK293T biosensor cells with plasmids to transiently overexpress HTRA1 and HTRA1$^{S328A}$. Cells treated with PFFs showed robust seeding, while cells overexpressing HTRA1 or HTRA1$^{S328A}$ that were subsequently treated with PFFs showed an apparent decrease in puncta accumulation, particularly in regions where transfection efficiency was higher (Fig. 8a). We transfected these constructs at two different levels to monitor any dose dependence and

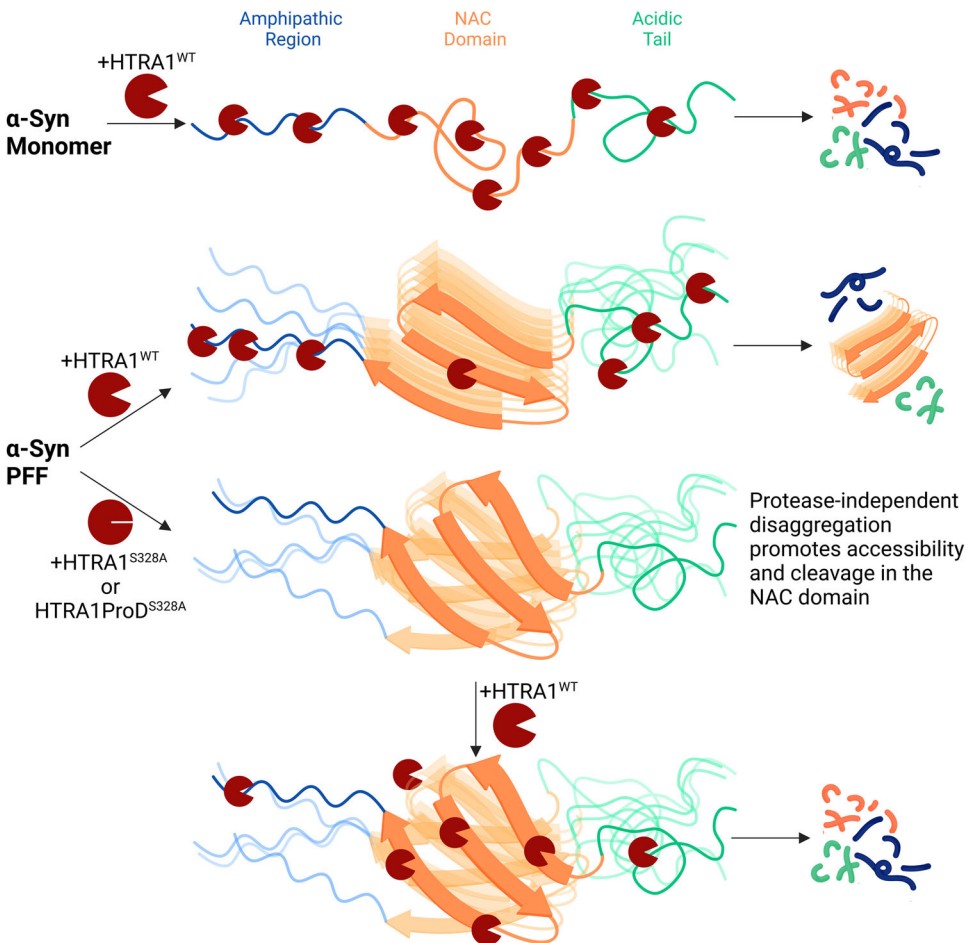

**Fig. 7 | Potential model of HTRA1-mediated disaggregation of α-synuclein.** HTRA1^WT can ordinarily cleave α-syn monomer throughout the α-syn sequence. Once α-syn forms fibrils (middle row), HTRA1 can still proteolyze α-syn, however these digestion sites are restricted to the N- and C-terminal regions as indicated by mass spectrometry data, Fig. 6a, d, where α-syn is more accessible. However, the NAC domain remains protected from cleavage. HTRA1^S328A or HTRA1ProD^S328A pre-treatment promotes disaggregation by favorably binding the fibrillar form of α-syn as indicated by pull-down data, Fig. 6f, g, and upon subsequent addition of HTRA1^WT (bottom row) proteolysis can occur throughout α-syn, including in the NAC domain.

quantified these effects by flow cytometry in the total population of cells (Fig. 8b, c and Supplementary Fig. 8A, B). Both HTRA1 and HTRA1^S328A decreased seeding by ~40%, with a dose-dependent increase in inhibitory activity at higher HTRA expression levels. When analyzing just the population of cells expressing the HTRA-Myc constructs, we observe similar effects (Supplementary Fig. 8F–I). Thus we conclude that in the cellular environment, HTRA1 can protect against α-syn seeding, and pre-treatment with HTRA1 is not required.

Because HTRA1 is expressed both in the cytoplasm and is secreted, these activities could be due to HTRA1 inhibiting α-syn aggregation in the cell, or following secretion, extracellular HTRA1 might prevent uptake of α-syn fibrillar seeds. To test these possibilities, we designed an N-terminal truncation construct (HTRA1ΔNTD) to prevent HTRA1 secretion[46]. We confirmed expression of all constructs (Fig. 8d), and that HTRA1ΔNTD was no longer secreted (Fig. 8e). Further, we find that α-syn PFFs are taken up similarly in cells expressing HTRA1^WT and HTRA1ΔNTD (Fig. 8f and Supplementary Fig. 8C) and note a stronger reduction in seeding by PFFs in cells overexpressing the HTRA1^ΔNTD construct than by HTRA1^WT (Fig. 8g and Supplementary Fig. S8D). Taken together, our result suggests that HTRA1 is primarily inhibiting α-syn aggregation intracellularly, and secretion of HTRA1 is not required for inhibition of seeding.

Finally, to investigate if HTRA1 plays a protective role against α-syn seeding, we transfected the biosensor cells with HTRA1 siRNA to knockdown endogenous HTRA1 expression levels. Upon subsequent treatment of the cells with PFFs, we observed an increase in seeding of ~40% as compared to the negative control siRNA-treated cells (Fig. 8h, i and Supplementary Fig. 8E). We therefore conclude that endogenous levels of HTRA1 can mitigate α-syn aggregation in HEK293T cells, and HTRA1 may have a native protective role against α-syn aggregation.

## HTRA1 treatment renders α-synuclein non-toxic and incompetent of seeding formation of pathological α-syn inclusions in primary mouse neurons

To evaluate the effect of HTRA1 treatment on α-syn PFF-induced seeding in mouse primary neurons, we incubated monomeric α-syn alone or with HTRA1, HTRA1^S328A, HTRA2, or aldolase and applied the products to primary mouse hippocampal neurons. It has been shown that α-syn PFFs can be taken up by neurons, seed, and convert soluble α-syn into Lewy Body-like inclusions, which are also associated with hyperphosphorylation of α-syn[5,39]. 24 h following treatment, toxicity was measured by MTT assay, and 1 week following treatment the neurons were processed for phosphorylated α-syn by ICC and imaged by confocal microscopy (Fig. 9a). Application of untreated PFFs decreases neuronal viability, and only ~70% of neurons remained viable. However, pre-treatment of α-syn with HTRA1 or HTRA1^S328A was partially protective, restoring viability to ~85% and 81%, respectively. This viability level is similar to that achieved when nontoxic monomeric α-syn is applied. Treatment with HTRA2 or aldolase did not modulate viability in a statistically significant manner (Fig. 9b).

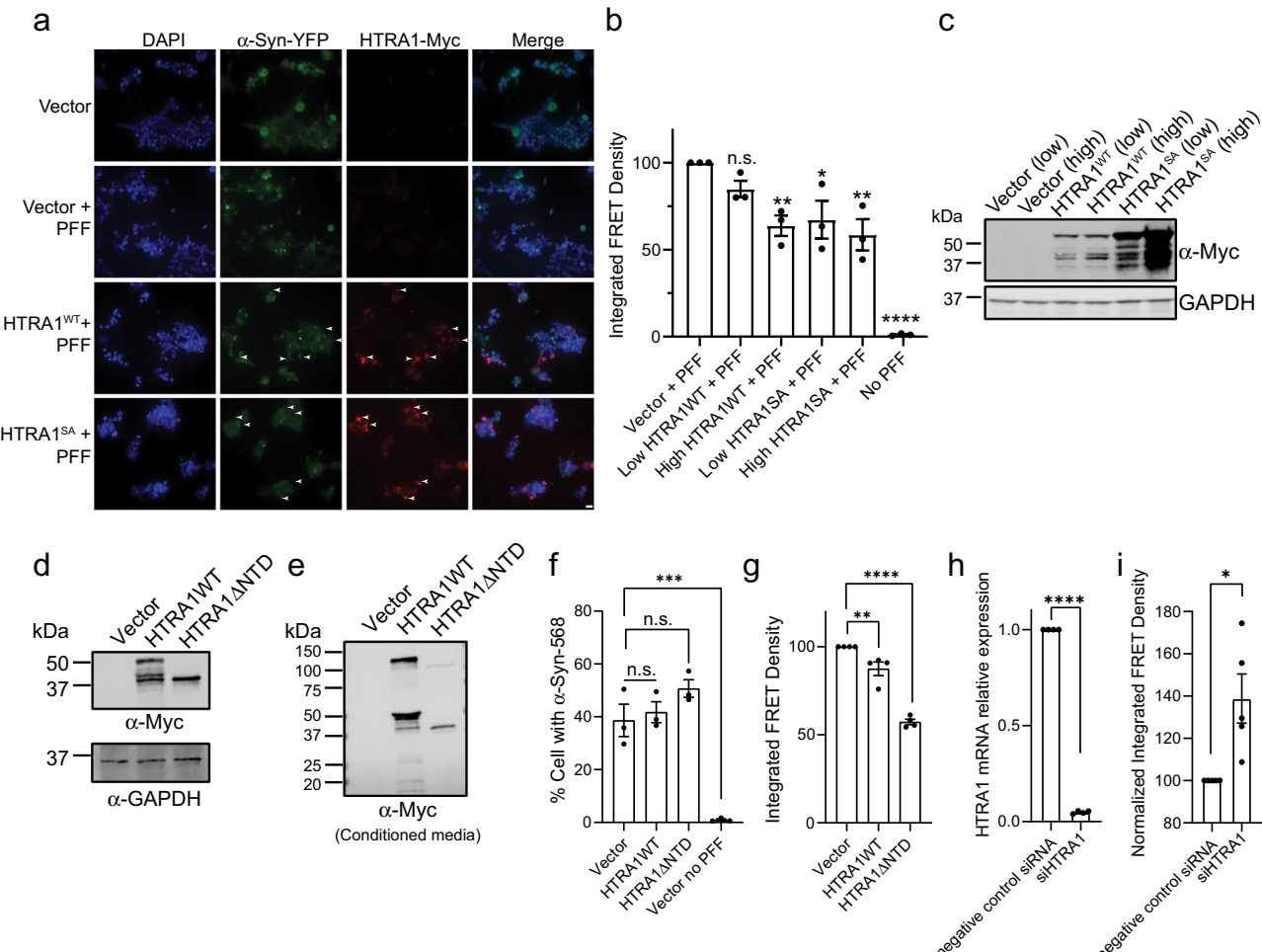

**Fig. 8 | Overexpression of HTRA1 prevents α-syn PFFs from seeding aggregation. a** HEK293T biosensor cells were transfected with HTRA1-Myc, HTRA1[S328A]-Myc plasmid, or vector. 2 days following transfection, α-syn PFFs were added. One day after treatment, cells were fixed and stained for DAPI (blue) or Myc (HTRA1, red). α-Syn was imaged via the YFP tag. Arrowheads indicate areas with high HTRA expression and low α-syn puncta formation. Scale bar, 20 µm. **b** HEK293T biosensor cells were transfected and analyzed by flow cytometry (N = 3 independent experiments, biological replicates are shown as dots, bars represent means ± SEM, respectively. *p = 0.0165, **p = 0.0086 (high HTRA1WT), **p = 0.0032 (high HTRA1SA), ****p < 0.0001 by one-way ANOVA with Dunnett's multiple comparisons test as compared to Vector + PFF). Analysis shows total cell population. **c** Cells from **b** were immunoblotted (N = 3 independent experiments). **d** HEK293T biosensor cells were transfected with HTRA1-Myc, HTRA1[ΔNTD]-Myc, or vector control and immunoblotted (N = 2 independent experiments). **e** Cell culture media from **d** was collected and immunoblotted (N = 1). **f** Biosensor cells were transfected with the indicated plasmid. 2 days following transfection, Alexa-568 labeled α-syn PFFs were added. Cellular internalization of Alexa-568 labeled α-syn PFFs was quantified by Alexa-568 signal. N = 3 independent experiments, bars represent means ± SEM, ***p = 0.0004 by one-way ANOVA with Dunnett's multiple comparisons test compared to vector + PFF. **g** HEK293T biosensor cells were transfected with indicated plasmids, treated with unlabeled α-syn PFFs, and analyzed by flow cytometry. N = 4 independent experiments, bars represent means ± SEM, **p = 0.0095, ****p < 0.0001 by one-way ANOVA with Dunnett's multiple comparisons test, values compared to Vector + PFF. **h** HEK293T biosensor cells were transfected with HTRA1 siRNA or a negative control. 2 days following transfection, α-syn PFFs were added to media (10 nM). Cells were then analyzed by RT-qPCR. N = 4 independent experiments, ****p < 0.0001 by two-sided unpaired t-test compared to negative control siRNA. **i** Cells from **h** were analyzed by flow cytometry. N = 5 independent experiments, bars represent means ± SEM, *p = 0.0103 by two-sided unpaired t-test compared directly to negative control siRNA.

To confirm that the mechanism of toxicity suppression is due to decreased seeding of intracellular α-syn, we next tested whether HTRA1 affected PFF-induced aggregation and hyperphosphorylation of α-syn. Here, reactions were prepared as described previously and neurons were treated for 1 week with the reaction products. Cells were then processed and immunostained for phosphorylated α-syn (Fig. 9c). Here, upon transduction of α-syn PFFs formed in the absence of HTRA1, we observe the accumulation of Lewy Body-like inclusions comprised of hyperphosphorylated α-syn in the cytosol and mislocalization of α-syn to the axons. Phosphorylated α-syn is a highly specific marker of α-syn pathology[47], and we can confirm that these inclusions are comprised of endogenous α-syn because the recombinant PFFs were not phosphorylated prior to transduction. Transduction of products formed in the presence of HTRA2 or aldolase induced

similar accumulation of hyperphosphorylated inclusions. However, when products were formed in the presence of HTRA1 or HTRA1[S328A], we observed no accumulation of phosphorylated α-syn inclusions. Thus we conclude that treatment with HTRA1 renders α-syn seeds nontoxic in neurons. These products are also incapable of seeding endogenous α-syn and restrict the formation of pathological, hyperphosphorylated Lewy Body-like inclusions of α-syn.

## Discussion

Here, we establish that HTRA1 can both prevent the amyloidogenesis of α-synuclein and dissolve preformed α-syn amyloid fibrils, independent of proteolytic activity. HTRA1 restricts monomeric α-syn from forming amyloid species and renders it incapable of seeding the aggregation of endogenous α-syn. Though HTRA1 is expressed

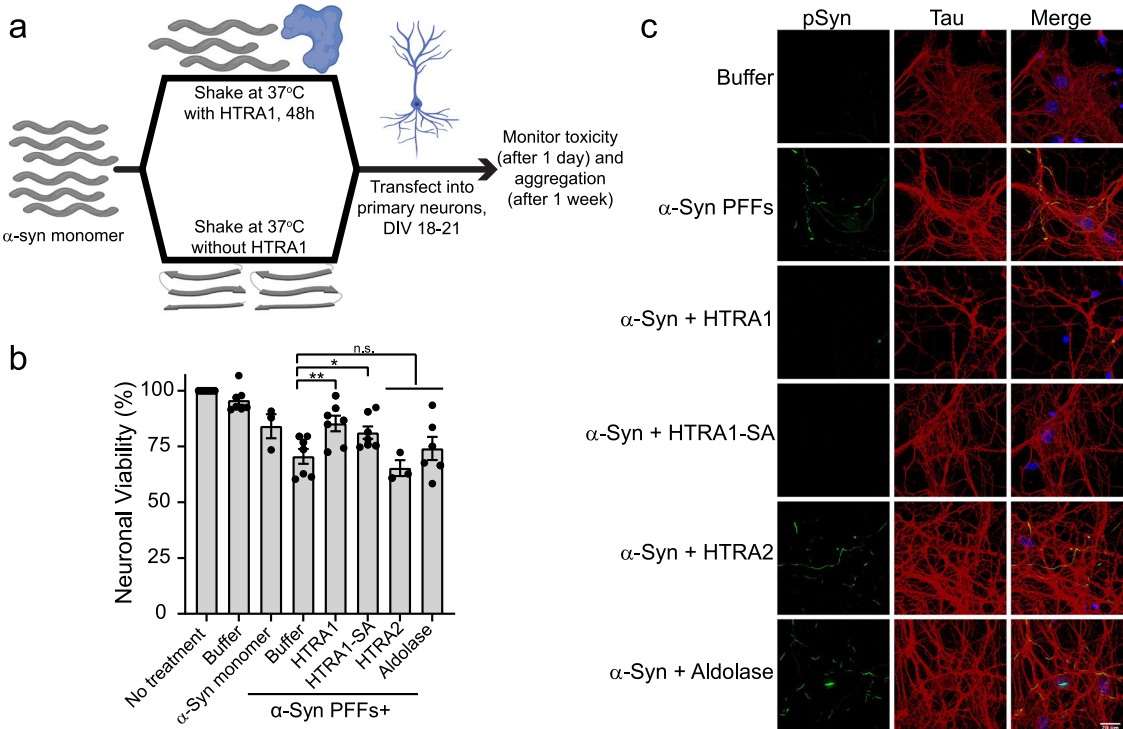

**Fig. 9 | HTRA1 treatment renders α-Syn non-toxic and seeding incompetent in primary mouse neurons. a** Schematic showing experimental design. **b** Fibrillization of α-syn (100 μM) was conducted by shaking for 48 h at 37 °C in the presence of HTRA1, HTRA1$^{S328A}$, HTRA2, buffer, or Aldolase (20 μM). Reaction products were applied to mouse primary hippocampal neurons at DIV 18–21. 24 h following treatment, cell viability was assessed by MTT assay. Each value is compared directly to α-syn + no treatment control using a series of two-sided Welch's *t*-tests (*N* = 3 for α-syn monomer and HTRA2; *N* = 6 for aldolase; *N* = 7 for buffer, α-syn PFFs + buffer, HTRA1, and HTRA1-SA; *N* = 8 for no treatment, biological replicates are shown as dots, bars represent means ± SEM, **p* = 0.031, ***p* = 0.0097). **c** Immunocytochemistry of neurons as treated in part **b**, 1 week following addition of α-syn/HTRA reactions. Endogenous phosphorylated-α-syn (green), tau (red), DAPI (blue). Scale bar, 20 μm.

intracellularly and is secreted, we find that it is primarily the intracellular form of HTRA1 that protects against α-syn seeding, and decreased HTRA1 expression levels make cells more susceptible to α-syn seeding. Furthermore, HTRA1 treatment prevents α-syn from forming neurotoxic species and also prevents formation of the characteristic α-syn amyloid conformers that drive aggregation of phosphorylated α-syn inclusions in neurons. This activity is not limited to the prevention of amyloidogenesis, as HTRA1 can also solubilize preformed α-syn amyloid fibrils, thereby decreasing their seeding capacity. Most known protein quality control systems are comprised of multicomponent machines that require ATP hydrolysis for remodeling[15]. In contrast, HTRA1 remodeling does not require collaboration with other proteins or ATP hydrolysis. This ATP-independent disaggregase activity is not unique to HTRA1, and has also recently been discovered in DAXX[23], TRIM proteins[25], and nuclear-import receptors such as karyopherin-β2 (Kapβ2)[24].

We find that HTRA1-mediated disaggregase activity is independent of HTRA1 proteolytic activity, and that disaggregation can be facilitated via the protease domain alone, suggesting that HTRA proteins leverage distinct mechanisms under different circumstances. Further, HTRA1 and HTRA2 appear to operate via distinct mechanisms. Mitochondrial dysfunction is implicated in PD[36], and inactivating mutations in the mitochondrial HTRA2 gene are associated with PD risk[26,33]. Evidence has shown that α-syn accumulates in the mitochondria in PD, which reduces mitochondrial activity. Thus we initially hypothesized that HTRA2 might be the principal HTRA isoform that mediates α-syn disaggregation. Furthermore, because α-syn fibrillar seeds would first enter the cytoplasm, HTRA2 might also be relevant to cytoplasmic α-syn. However, although we find that HTRA2 binds α-syn monomer with similar affinity to HTRA1$^{S328A}$, HTRA2 has only limited

remodeling activity against α-syn. These results suggest that physical association alone does not account for the remodeling conferred by HTRA1 in detoxifying α-syn. Similar phenomena were described for Kapβ2, which disaggregates FUS[24]. In those studies, an antibody that binds FUS in the same region as Kapβ2 did not mediate disaggregation of FUS fibrils. With HTRA1, our findings give further support to the concept that disaggregation and proteolysis operate via distinct mechanisms to regulate proteopathic aggregates. In future studies it will be important to further explore these differences. Additional studies could also further explore potential mitochondria-specific roles of HTRA2.

Amyloid species are generally characterized as protease-resistant, and so we aimed to better understand how HTRA1 dissolves these species. We demonstrate that HTRA1 disaggregase activity can function in a proteolytically-independent fashion to solubilize otherwise recalcitrant species. Surprisingly, the proteolytic activity of HTRA1 appears to restrict its remodeling activity, as the HTRA1$^{S328A}$ protease-inactive mutant was more protective in most settings. Further, while PDZ domains typically mediate protein-protein interactions and bind β-sheet rich proteins through a β-sheet augmentation mechanism[27], we surprisingly find that the PDZ domain is not required for mediating the HTRA1−α-syn binding interaction and that the PDZ domain is dispensable for disaggregase activity. Instead, the protease domain binds substrate, mediates remodeling and, when active, can proteolyze the product.

To further explore this mechanism, we probed these reactions using mass spectrometry. We find that HTRA1 ordinarily cleaves monomeric α-syn at many different positions throughout the α-syn sequence. However, upon fibrillization, the NAC domain of α-syn PFFs becomes inaccessible to HTRA1 proteolysis, with no cleavage

occurring in the NAC domain. Instead, cleavage was restricted to the N- and C-terminal regions. However, upon addition of HTRA1[S328A], we observe several new cleavage sites are present, and these sites are particularly enriched in the NAC domain. Similar results are achieved for HTRA1ProD[S328A], providing additional evidence that the PDZ domain is dispensable for these activities. Furthermore, we find that HTRA1ProD[SA] binds α-syn PFFs with greater affinity than monomer (Fig. 6f, g). This suggests that disaggregation is mediated by engagement of the aggregated fibrillar form of α-syn. Thus, we propose a mechanism whereby HTRA1 ordinarily mediates cleavage of α-syn monomer in the NAC domain. Upon fibrillization, HTRA1 mediates solubilization of the α-syn NAC domain via its protease domain, thereby promoting disaggregation. Once the NAC domain is solubilized, it becomes susceptible to proteolysis (Fig. 7). These findings are contrary to previous studies suggesting that the PDZ domain is essential for substrate binding by HTRA1[28]. However, in this earlier study by Poepsel et al., the authors demonstrated that a PDZ domain deletion variant still maintained high proteolytic activity, with just subtle impairment as compared to the full-length protein, suggesting that the PDZ domain is instead dispensable. Further, they found that the HTRA1 PDZ domain alone could not solubilize tau fibrils[28], while we find that the HTRA1 protease domain alone retains activity against α-syn. In future studies, it will be important to better understand these features and how HTRA1 operates via a distinct mechanism as compared to canonical proteases. It will be important to fully elucidate these mechanisms, perhaps by quantitative comparison of binding of different HTRA1 species to different α-syn species. Further, the specific way in which HTRA1 engages and solubilizes α-syn remains unclear. Our data suggests that HTRA1 mediates solubilization via the NAC domain of α-syn, and we propose that HTRA1 does so through direct binding of the NAC domain, however we cannot exclude the possibility that HTRA1 binds another region of α-syn and remodels the NAC domain allosterically. A comprehensive understanding of this mechanism will be an important line of future investigation.

We find that HTRA1 can also inhibit amyloidogenesis of α-syn[A53T], which proceeds more rapidly and is associated with early-onset PD[41]. Beyond the remodeling activity of HTRA1 against α-syn and tau, we find that HTRA1 can also proteolyze and inhibit the aggregation of the amyloid-like proteins TDP-43 and FUS. However, HTRA1 appears tuned to the properties of amyloid, as its activity is more potent against α-syn than TDP-43 or FUS, and it does not proteolyze well-folded proteins including GST and MBP. Furthermore, we find that PDZ domain binding is similarly dispensable for HTRA1-mediated inhibition of TDP-43 and FUS aggregation. In the future, it may be possible to engineer HTRA1 variants with enhanced disaggregation activity against α-syn[A53T] and other substrates implicated in neurodegeneration. Such a strategy has been successful with the yeast amyloid disaggregase Hsp104, which has been engineered to counter the misfolding of α-syn, TDP-43, FUS, and other disease-associated proteins[48–54].

When proteins form amyloid, this is typically viewed as an irreversible conversion, whereby the proteins become resistant to processing and clearance from the cell. This is problematic due to the possible loss of function of the misfolded protein, as well as the accumulation of potentially toxic species. Our work suggests that HTRA1 may function as a chaperone that maintains proteostasis by regulating the folding of α-syn. Although we find that intracellular HTRA1 provides crucial protection against α-syn seeding, because HTRA1 is also secreted, it may also function in preventing cell-to-cell propagation of amyloid seeds. However, it is unclear whether deficits in HTRA1-mediated disaggregase activity contribute to pathologic α-syn aggregation in PD. In developing new therapeutic strategies for neurodegeneration, it is possible that small molecule therapeutics will be insufficient to prevent or actively clear accumulations of amyloid and amyloid-like misfolded species. Disaggregases are a promising alternative therapeutic approach. They have the capacity to engage

and remodel misfolded and amyloid species, countering both a possible loss of function or gain of toxic function[15]. Modulation of HTRA1 is particularly intriguing in this regard, as it could be harnessed to dissolve misfolded species and allow for either their reactivation or degradation. However, due to the broad substrate repertoire of HTRA1, therapeutic strategies targeting HTRA1 will need to be carefully designed. Simply increasing HTRA1 expression levels may be problematic, as this may lead to proteolytic degradation of otherwise functional proteins. Instead, our study suggests that generation of smaller constructs, such as variants of HTRA1ProD[S328A], may allow for disaggregation without proteolysis of the native substrates of HTRA1. In the future, the development of small molecule modulators of HTRA1 or the engineering of HTRA1 variants with such properties could be a promising new avenue for the development of tailored therapeutics to modulate protein quality control[14,55].

## Methods

### Ethical statement
Animal procedures were performed in accordance with protocols approved by the Institutional Animal Care and Use Committee at Washington University School of Medicine.

### Oligonucleotides
All oligonucleotides used are listed in Supplementary Table 1.

### Protein purification
All proteins were expressed and purified from *E. coli* BL21-Codon-Plus(DE3)-RIL cells (Agilent) and purified under native conditions unless otherwise noted. Plasmid containing the HTRA1ΔNTD (residues 156-480) with a C-terminal 6-His tag in the pET21a plasmid was obtained from the Saghatelian lab[56]. The constructs: HTRA1ΔNTD[S328A], HTRA1ProD (residues 156-379), HTRA1PDZ (residues 380-480), and HTRA1ProDSA (residues 156-379) were generated by site-directed mutagenesis, with sequences confirmed by Sanger sequencing. The HTRA2ΔNTD (residues 134-458) gene with a C-terminal 6-His tag in the pET21d plasmid was obtained from Genscript. To generate recombinant protein, *E. coli* cells were induced at $OD_{600} = 0.6$ with 0.4 mM IPTG for 18 h at 16 °C. Cell pellets were resuspended in HTRA wash buffer (50 mM Tris, pH 8.0, 1 M NaCl, and 30 mM imidazole) supplemented with lysozyme (20 mg per L of initial culture) and protease inhibitors (cOmplete, EDTA free, Roche). Cells were lysed by sonication and the lysate was cleared by centrifugation. The supernatant was then incubated with Fast flow nickel sepharose (GE Healthcare) for 2 h at 4 °C, and all subsequent steps occurred at 4 °C. The resin was then transferred to a column, washed with HTRA wash buffer, and eluted with HTRA elution buffer (50 mM Tris, pH 8.0, 100 mM NaCl, and 500 mM imidazole). The protein was then buffer exchanged into HTRA storage buffer (50 mM Tris, pH 8.0, 100 mM NaCl, 10% glycerol) and concentrated to ~5–10 mg/mL. The protein was then flash frozen in liquid nitrogen before storage at −80 °C. Protein was stored in the freezer for no longer than three months to minimize auto-proteolysis.

Plasmids for expression of α-synuclein were from Peter Lansbury[57]. α-Synuclein was purified as described[39]. Briefly, α-syn was expressed in *E. coli* BL21-DE3-RIL cells (Invitrogen), where expression was induced at $OD_{600} = 0.6$ with 1 mM IPTG for 2 h at 37 °C. Cell pellets were resuspended in osmotic shock buffer (30 mM Tris, pH 7.2, 2 mM EDTA, 40% sucrose). Cells were lysed by incubating in osmotic shock buffer for 10 min at room temperature, centrifuged, and resuspended in 0.84 mM MgCl₂. Lysate was then cleared by centrifugation. Nucleic acids were removed via streptomycin sulfate precipitation. The supernatant was then boiled for 10 min, after which most proteins precipitate while α-syn remains soluble following boiling. Protein was then loaded onto a bed of DEAE sepharose for anion-exchange. The column was washed with wash buffer (20 mM Tris, pH 8, 1 mM EDTA) and eluted with elution buffer (20 mM Tris, pH 8.0, 300 mM NaCl,

1 mM EDTA). The eluate was then dialyzed into α-syn fibrillization buffer (20 mM Tris, pH 8.0, 100 mM NaCl), flash frozen, and stored at −80 °C until use.

GST-FUS was expressed and purified as described[38]. Briefly, GST-FUS was expressed in *E. coli* and expression was induced at $OD_{600} = 0.6$ with addition of 0.5 mM IPTG for 18 h at 16 °C. Cell pellets were resuspended in FUS wash buffer (PBS supplemented with 2 mM DTT, 100 μM PMSF, 10 μM pepstatin A, and cOmplete protease inhibitors) supplemented with lysozyme. Lysis was completed with sonication or homogenization and lysates were cleared by centrifugation. The lysate was then incubated with Glutathione Sepharose Fast Flow (GE Healthcare) resin for 1 h at 4 °C, washed with FUS wash buffer, and eluted with FUS elution buffer (50 mM Tris, pH 8.0, 200 mM trehalose, and 20 mM glutathione). Protein was then flash frozen and stored at −80 °C until use.

A TDP-43-MBP-His$_6$ construct was obtained from Addgene and purified as described[58]. Briefly, TDP-43-MBP was expressed in *E. coli* and expression was induced at $OD_{600} = 0.5$ with addition of 1 mM IPTG for 18 h at 16 °C. Cell pellets were resuspended in TDP-43 lysis buffer (50 mM HEPES, pH 7.4, 0.5 M NaCl, 30 mM imidazole, 10% glycerol, 2 mM DTT, 100 μM PMSF, 10 μM pepstatin A, and cOmplete EDTA-free protease inhibitors) supplemented with lysozyme. Lysis was completed by sonication and lysates were cleared by centrifugation. The lysate was then incubated with Fast flow nickel sepharose (GE Healthcare) for 1–1.5 h at 4 °C. The resin was then washed with TDP-43 wash buffer (50 mM HEPES, pH 7.4, 0.5 M NaCl, 30 mM imidazole, 10% glycerol, 2 mM DTT) and eluted in buffer (50 mM HEPES, pH 7.4, 0.5 M NaCl, 0.5 mM imidazole, 10% glycerol, 2 mM DTT, 100 μM PMSF, 10 μM pepstatin A, and cOmplete protease inhibitors). The eluent was then added to amylose resin (NEB) and incubated for 90 min at 4 °C, washed with amylose wash buffer (50 mM HEPES, pH 7.4, 0.5 M NaCl, 10% glycerol, 2 mM DTT), and eluted with amylose elution buffer (50 mM HEPES, pH 7.4, 0.5 M NaCl, 10% glycerol, 2 mM DTT, 100 μM PMSF, 10 μM pepstatin, 10 mM maltose, and cOmplete protease inhibitors). The eluent was concentrated to ~40 μM using a 30 kDa molecular weight cutoff filter, flash frozen, and stored at −80 °C until use.

### Proteolysis assays

α-Syn monomer or fibril (25 μM) was treated with buffer or the indicated HTRA construct (5 μM) for the indicated time at 37 °C. Samples were then processed by SDS-PAGE. GST-FUS (10 μM) or TDP-43-MBP-His$_6$ (10 μM) were treated with TEV protease for 1 h at 37 °C to liberate free FUS and TDP-43. Following cleavage, buffer or the indicated HTRA construct (2 μM) was added for 24 h at 37 °C. For GST and casein proteolysis, purified GST (25 μM) was treated with buffer or the indicated HTRA construct (5 μM) for 24 h at 37 °C. Casein (40 μM) was treated with buffer or the indicated HTRA construct (2.5 μM) for 24 h at 37 °C. Samples were then processed by SDS-PAGE. For assays monitoring degradation of FITC-tagged casein, FITC-casein (10 μM) was treated with HTRA1, HTRA1ProD, or HTRA1PDZ (2 μM) for the indicated time at room temperature. Degradation of FITC-Casein was monitored at 482 nm after excitation at 450 nm using a Tecan Spark plate reader. Data acquisition was performed with Tecan SparkControl software. All uncropped gels and blots are shown in the Source Data file.

### Inhibition of FUS and TDP-43 aggregation

FUS aggregation reactions were prepared by mixing GST-TEV-FUS in FUS assembly buffer (50 mM HEPES, pH 7.4, 10% glycerol, 1 mM DTT) supplemented with 1 mM DTT with HTRA constructs or buffer control at the indicated concentrations. Reactions were initiated by addition of TEV protease and reactions were monitored for turbidity by continuously measuring absorbance at 395 nm at 25 °C without agitation

in a BioTek Epoch plate reader, and data acquisition was performed using BioTek Gen 5 software.

TDP-43 aggregation reactions were prepared in a similar way in TDP-43 assembly buffer (50 mM HEPES, pH 7.4, 10% glycerol, 1 mM DTT). Here TDP-43-TEV-MBP was mixed with the indicated HTRA construct or buffer control at the indicated concentrations. Reactions were initiated by addition of TEV protease and reactions were monitored for turbidity by continuously measuring absorbance at 395 nm at 30 °C with agitation in a BioTek Epoch plate reader, and data acquisition was performed using BioTek Gen 5 software.

### Preparation of α-Syn preformed fibrils (PFFs)

To prepare α-syn preformed fibrils, monomeric α-syn was filtered through a 0.2 μM filter. Monomer (5 mg/mL) was then diluted in fibrillization buffer (20 mM Tris, pH 8, 100 mM NaCl) and incubated at 37 °C with agitation at 1500 rpm in an Eppendorf Thermomixer for 7 days to produce mature fibrils. The resulting mixture was centrifuged at 21,130×*g* for 30 min at room temperature. The supernatant was then removed and a BCA assay was used to determine the concentration of the fibrils. PFFs were resuspended in fibrillization buffer to 5 mg/mL.

Preparation of Alexa-568 labeled α-syn was performed as previously described[59]. Briefly, 5 mg/mL monomeric α-syn was filtered through a 0.2 μm syringe filter and diluted to 2 mg/mL in 0.1 M NaHCO$_3$. Alexa Fluor 568 NHS Ester (A20003) was dissolved in DMSO to 10 mg/mL. Alexa dye was added in 2.1:1 molar ratio of dye:α-syn and mixed by stirring at room temperature for 1 h. Unbound dye was removed using Bio-Spin P-6 Gel Columns (Bio-Rad #7326227). The Alexa-568 labeled α-syn was aliquoted, flash frozen, and stored at −80 °C. To prepare labeled α-syn PFFs, both monomeric Alexa-568 labeled α-syn and unlabeled α-syn were filtered through a 0.2 μm syringe filter. Alexa-568-α-syn was mixed with unlabeled α-syn (5% labeled) in 40 mM HEPES pH 7.4, 150 mM KCl, and 20 mM MgCl$_2$. The mixture was then incubated at 37 °C with agitation at 1,500 rpm in an Eppendorf ThermoMixer with ThermoTop for 7 days to produce mature fibrils. The resulting mixture was centrifuged at 21,130×*g* for 30 min at room temperature. The supernatant was then removed and PFFs were resuspended in 20 mM Tris pH 8, 100 mM NaCl to achieve 5 mg/mL fibrils. Protein was then aliquoted, flash frozen and stored at −80 °C until use.

### α-Syn inhibition and disassembly reactions

For inhibition reactions, α-synuclein monomer (25 μM) was incubated in fibrillization buffer (20 mM Tris, pH 8.0, 100 mM NaCl) with or without the indicated HTRA construct or aldolase (5 μM) at 37 °C with agitation at 1500 rpm in an Eppendorf Thermomixer.

For disassembly reactions, α-syn PFFs (10 μM) were incubated in fibrillization buffer with or without HTRA1, HTRA1$^{S328A}$, or aldolase (100 μM) at 37 °C with gentle shaking at 350 rpm in an Eppendorf Thermomixer for 24 h.

Fibril assembly and disassembly was monitored by ThioflavinT (ThT) fluorescence. Here, ThT (10 μM) was mixed with α-synuclein fibrils (0.5 μM). Fluorescence at 482 nm was measured after excitation at 450 nm using a Tecan Spark plate reader.

### Sedimentation assays

To monitor inhibition of α-syn fibrillization, reactions were prepared as above were taken at the indicated time points. Reactions were then centrifuged at 21,130 x g for 30 min at room temperature to separate the soluble and insoluble fractions. Following centrifugation, the supernatant and pellet were resuspended in sample buffer (60 mM Tris, pH 6.8, 5% glycerol, 2% SDS, 4% β-mercaptoethanol). The total, soluble, and pellet fractions were then boiled, resolved by SDS-PAGE, and stained with Coomassie Brilliant Blue. For TDP-43 and FUS, sample preparation was performed as described above.

To monitor PFF disassembly, α-syn PFFs (5 μM) were incubated in fibrillization buffer with the indicated HTRA1 variant (100 μM) at 37 °C for 48 h. The total, soluble, and pellet fractions were then spotted on nitrocellulose membranes and probed with anti-syn1 antibody (BD Science, Cat 610787, 1:1000 dilution).

Image acquisition was performed using Bio-Rad Image Lab 6.0.1 software. The amount in either fraction was determined by densitometry using the Image Lab 6.0.1 software on a Bio-Rad Gel Doc EZ Imaging system.

## Electron microscopy
Samples of α-synuclein incubated with or without HTRA1 as described above were applied to 200 mesh, pure carbon, copper grids (Ted Pella #0184-F). The grids were then washed with water five times, and stained with 2% uranyl acetate for 1 min.

Images were obtained using a JEOL JEM-1400 120 kV transmission electron microscope.

## Pull down assay
Interactions between HTRA constructs and α-synuclein monomer or fibril were examined by His-mediated pull-down assays. Recombinant HTRA-6His (0.15 mg) was immobilized to 50 μL of Ni-Sepharose resin (GE Healthcare Cytiva, cat: 45002985) and then incubated with 0.15 mg of wild-type α-synuclein at room temperature in assay wash buffer (20 mM Tris, 100 mM NaCl, 10 mM Imidazole, pH 8.0). The incubated mixture was washed five times with wash buffer, and eluted with 500 mM imidazole. Protein samples were collected prior to the wash step as 'Input'. All samples were then boiled in sample buffer (60 mM Tris, pH 6.8, 5% glycerol, 2% SDS, 4% β-mercaptoethanol) and resolved by SDS-PAGE. For Western blot analysis, proteins were transferred to nitrocellulose membrane and probed with anti-syn1 antibody (BD Science, Cat 610787, 1:1000 dilution). Membranes were imaged using a Li-COR Odyssey FC Imaging system and the amount of protein in 'input' and 'bound' fractions was determined by densitometry using Image Studio Lite 5.5.4 software.

## Fluorescence polarization
An α-synuclein-Alexa-488 construct was prepared with dye introduced at a single site. The S9C mutation was introduced into the α-syn sequence by site-directed mutagenesis. S9C-α-syn monomer was purified as described above. Following DEAE-sepharose anion exchange chromatography, monomer was labeled with Alexa Fluor 488 using a C5-maleimide reagent (Invitrogen).

HTRA constructs or BSA (Thermo Scientific, Cat #23209) were serially diluted in polarization buffer (20 mM Tris, pH 8.0, 100 mM NaCl, 5% glycerol, 5% sucrose). Alexa-488 labeled α-syn monomer was used at a final concentration of 20 nM, mixed with protein preparations, and added to a 384-well black flat bottom plate. The protein mixtures were incubated at room temperature for 1.5 h, protected from light. Fluorescence polarization was measured at 520 nm after excitation at 470 nm using a Tecan Spark plate reader.

## Native-PAGE analysis
For native-PAGE analysis of protein complex formation, inhibition reactions following 48 h incubation were prepared in native-PAGE sample buffer (62.5 mM Tris-HCl, 40% glycerol, pH 6.75) at a 1:1 ratio. Protein complexes were then separated on a 4-20% non-denaturing gradient polyacrylamide gel and stained with Coomassie Brilliant Blue.

For MS analysis, the band was excised from the gel, followed by in-gel digestion. Proteomics was performed by MS Bioworks (Ann Arbor, MI). The digested sample was analyzed by nano LC-MS/MS with a Waters M-Class HPLC system interfaced to a ThermoFisher Fusion mass spectrometer. Protein identification was performed using Mascot (Matrix Science). Protein validation and visualization was then performed using Scaffold (Proteome Software). The mass spectrometry proteomics data have been deposited to the ProteomeXchange consortium via the JPOST repository[60] with the dataset identifiers: PXD044806 for ProteomeXchange and JPST002295 for jPOST.

## HEK293T cell culture
HEK293T biosensor cells were obtained from Tritia Yamasaki[61]. Cells were grown in Dulbecco's modified high glucose Eagle's medium (DMEM) supplemented with 10% fetal bovine serum (FBS), and 1% penicillin/streptomycin. Control cell lines, HEK293T, HEK293T-α-syn-CFP and HEK293T-α-syn-YFP lines were cultured in the same conditions. For FRET seeding assays, the biosensor cells (HEK293T-α-syn-CFP/α-syn-YFP) were plated in 96-well plates at a density of 35 K cells per well. Inhibition and disassembly reactions were prepared as described above and used following 48 h (inhibition) or 24 h (disassembly) treatment with HTRA or control. Samples were then sonicated in a cup horn water bath sonicator (QSonica) at 65amp for 3 min, packaged with 0.5 μL Lipofectamine 3000 (Invitrogen), and transduced into the biosensor cells. Here, 24 h following plating, treated samples were added dropwise to achieve a final concentration of α-syn of 50 nM (inhibition) or 10 nM (disassembly) in each well. Cells were then harvested after 48 h (inhibition) or 24 h (disassembly) and processed for flow cytometry analysis. For flow cytometry, cells were detached with 0.05% Trypsin/EDTA, followed by fixation with 4% paraformaldehyde for 15 min at 4 °C in the dark. For immunostaining, cells were permeablized with 0.1% TX100/3% BSA in PBS and incubated with primary antibody (anti-Myc, Cell Signaling CAT#2278, 1:100 dilution) for 1 h at 4 °C. Cells were then washed and incubated with secondary antibody (Alexa-568 CAT#A-11011, 1:200 dilution) 1 h at 4°C. Cells were then resuspended in MACSQuant Flow Running buffer for analysis in a MACSQuant VYB flow cytometer. Fluorescence compensation was performed with control cell lines (HEK293T-α-syn-CFP and HEK293T-α-syn-YFP) each time prior to sample analysis. Following excitation of the CFP donor fluorophore with a 405 nm laser, FRET signal was monitored from the YFP acceptor fluorophore at 525 nm with a 50 nm bandpass filter. All data analysis was performed with FlowJo V10 software to determine the percent of FRET-positive cells and median FRET fluorescence intensity for each sample. The percent of FRET-positive cells was then multiplied by the median FRET intensity to calculate integrated FRET density, which was then normalized to a vehicle control.

## HTRA transient transfection and α-Syn seeding assay
For transient expression of HTRA constructs, cells were plated in 6-well plates at 100,000 cells per cm². Plasmids containing HTRA1 or HTRA2 with C-terminal Myc-DDK tags in the pCMV6 vector were obtained from Origene. The constructs: HTRA1^S328A and HTRA1ΔNTD (residues 156-480) were generated by site-directed mutagenesis, with sequences confirmed by Sanger sequencing. Transfections were performed 16-24 h after plating, at 70% confluence, using Lipofectamine 3000 (Invitrogen, Carlsbad, CA). Following two days of HTRA expression, α-syn PFFs were transduced as described above. Following an additional day of PFF treatment, cells were harvested (3 days post-HTRA1 transfection, 1 day post PFF transduction). Populations of cells were then split into two fractions for flow cytometry analysis or immunoblotting. Flow cytometry sample preparation was performed as described above. For immunoblotting, cells were pelleted and lysed by vortexing in modified RIPA buffer (50 mM Tris-HCl, pH 7.4, 150 mM NaCl, 0.5% TX-100, 0.5% deoxycholate, cOmplete protease inhibitors). Crude lysates were then centrifuged at 1000×g for 10 min at 4 °C. Total protein was quantified by BCA assay and equal amounts of total protein from each sample were prepared in 1xLaemmli sample buffer and boiled for 5 min. Lysates were then separated by SDS-PAGE (4-20% gradient, BioRad) and transferred to a PVDF membrane. Membranes were blocked in Odyssey Blocking Buffer (LI-COR) for at least 1 h. For conditioned media experiments, culture media was collected and centrifuged at 16,000×g for 10 min at r.t. to remove cellular debris.

Each sample was then prepared in 1x Laemmli sample buffer and boiled for 5 min. Primary antibody incubations were performed at 4 °C overnight. Primary antibodies used: anti-Myc (Proteintech Cat No. 60003-2-Ig, 1:1000 dilution), anti-Myc (Cell Signaling CAT#2278, 1:500 dilution), anti-α-syn (BD Bioscience, CAT#610787, 1:1000 dilution), anti-GAPDH (Proteintech, CAT#10494-1-AP, 1:2500 dilution). Membranes were then incubated with 680RD anti-Rabbit IgG (LI-COR CAT#926-68071, 1:2500 dilution) and 800CW Goat anti-Mouse IgG (LI-COR CAT#926-32210, 1:5000 dilution). Membranes were imaged using a Li-COR Odyssey FC Imaging system.

For immunocytochemistry, cells were fixed in paraformaldehyde (4%) for 15 min, followed by permeabilization and blocking with 3% BSA/0.1% TX-100 for 15 min. Cells were then labeled with primary antibody at 4 °C overnight. Cells were washed with PBS and then incubated with secondary antibody (goat anti-mouse Alexa-488, CAT#A-1101, 1:1000 dilution/ goat anti-rabbit Alexa-568, CAT#A-11011, 1:1000 dilution or goat anti-mouse Alexa-568, CAT#A-11004, 1:1000 dilution) for 1 h at room temperature. Primary antibodies used: anti-pSyn (Abcam MJFR13, ab168381, 1:5000 dilution), anti-Tau (Millipore Sigma, CAT#MABN827, Clone T49, 1:2000 dilution), and anti-Myc (Proteintech Cat#60003-2-Ig, 1:1000 dilution). Nuclei were stained with DAPI for 5 min. Cells were mounted onto slides using with Prolong Gold mounting solution. Images were acquired using a Nikon Eclipse Te200-E microscope with a 20x Plan Apo objective, image acquisition was performed with NIS-Elements AR 3.2, and processed with ImageJ 2.0.0.

### HTRA1 siRNA Knockdown and α-Syn seeding assay

For siRNA knockdown of HTRA1, cells were plated in 6-well plates at 50,000 cells per cm². HTRA1 siRNA (s11279) was pre-designed by Life Technologies, and obtained from Thermo Silencer Select siRNA. Transfections of siRNA were performed 16-24 h after plating, using Lipofectamine RNAiMAX (Invitrogen, Carlsbad, CA) with 25pmol siRNA for each well. Following two days of siRNA transfection, α-syn PFFs were transduced as described above, at 10 nM final concentration. Following an additional day of PFF treatment, cells were harvested. Populations of cells were then split into two fractions for flow cytometry analysis or RT-qPCR. Flow cytometry sample preparation was performed as described above. For RT-qPCR, cells were pelleted and homogenized in TRIzol Reagent (ThermoFisher) for total RNA extraction, according to manufacturer instructions. RT-qPCR was performed with 40 ng total RNA using EXPRESS One-Step Superscript pRT-PCR kit (Life Technology), and reactions were carried out at 20 μl final volume. HTRA1 mRNA levels were probed with PrimeTime qPCR assays (Integrated DNA Technologies) using a probe against HPRT1 as an internal control. Samples were run on an Applied Biosystems StepOnePlus Real-Time PCR System with the following cycling conditions: 50 °C for 15 min (1 cycle); 95 °C for 2 min (1 cycle); 95 °C for 15 s and 60 °C for 1 min (40 cycles). Data acquisition was performed using Life Technologies StepOne software v 2.3. Comparative cycle threshold (Ct) values of HTRA1 and HPRT1 were used to determine relative mRNA expression of HTRA1. The following primer/probe sets were used, with a probe to primer ratio of 2:1:

HTRA1 (Assay Name: Hs.PT.58.45742522)

Probe: 5′-/56-FAM/CCAGACCGA/ZEN/CGCCATCATTCAACT/3IABkFq/-3′ (NM_002775); Primer 1: 5′-CGCAACTCAGACATGGACTA-3′; Primer 2: 5′-GGAGATTCCAGCTGTCACTT-3′

HPRT1 (Assay Name: Hs.PT.58 v.45621572)

Probe: 5′-/56-FAM/AGCCTAAGA/ZEN/TGAGAGTTCAAGTTGAGTTTGG-3IABkFQ/-3′ (NM_000194); Primer 1: 5′-TTGTTGTAGGATATGCCCTTGA-3′; Primer 2: 5′-GCGATGTCAATAGGACTCCAG-3′

### Peptide sample preparation for mass spectrometry

To prepare samples for mass spectrometry analysis, α-synuclein monomer (25 μM) was incubated with HTRA1$^{WT}$ (5 μM) at 37 °C with

shaking at 350 rpm for 3 hr in an Eppendorf Thermomixer. The samples were reduced with 5 mM DTT, centrifuged at 21,130 x g for 30 min at room temperature to clear any insoluble material, and then subjected to C18 desalting.

For analysis of the fibrillar proteolysis, α-syn PFFs (5 μM) were pre-treated with HTRA1$^{S328A}$, HTRA1ProD$^{S328A}$, or HTRA1PDZ (50 μM), or buffer (50 mM Tris, pH 8.0, 100 mM NaCl) at 37 °C with shaking at 350 rpm for 2 h, followed by the addition of HTRA1$^{WT}$ (2.5 μM). Samples were then incubated for an additional 3 h at 37 °C with shaking in an Eppendorf Thermomixer. Samples were then solubilized in 6 M urea supplemented with 10% formic acid at 60 °C for 30 min with shaking at 600 rpm. The samples were then reduced with 5 mM DTT, and centrifuged at 21,130×$g$ for 30 min at room temperature to clear any insoluble material. The supernatant was transferred to a fresh protein low-bind tube and desalted with a C18 spec tip (Varian, cat# A57203). After C18 desalting, samples were dried under speedvac and resuspended (0.1% formic acid, 3% acetonitrile) prior to LC/MS analysis.

### LC/MS analysis of α-synuclein peptides

Peptide mixtures were analyzed by LC/MS by using a UHPLC system coupled to an Orbitrap ID-X Tribrid mass spectrometer (Thermo Fisher Scientific). The following electrospray ionization conditions were used: sheath gas flow 32 arbitrary units (Arb), auxiliary gas flow 5 Arb, sweep gas flow 0 Arb, ion transfer tube temperature 325 °C, and vaporizer temperature 125 °C. The RF lens value was 60%. Data were acquired in positive polarity with a spray voltage of 3.5 kV. MS1 data were acquired at a resolution of 60 K with an automatic gain control (AGC) target of 4e5 and a maximum injection time of 100 ms. MS/MS spectra were collected on [M + H]+ ions in positive polarity for each sample by using DDA. The MS/MS isolation window was set to 1.6 $m/z$. A normalized collision energy (NCE) of 30% was used. MS/MS data were acquired with 15 K resolution, an AGC target of 1.25e4, a maximum injection time of 86 ms, and a dynamic exclusion of 10 s. The intensity threshold was set to 2.5e4. Samples were randomized before analysis. Negative control sample containing only α-synuclein fibrils were injected and analyzed to preclude the identification of α-synuclein fragments resulting from protein purification. In addition, a quality-control (QC) sample was injected to monitor signal stability of the instrument.

MaxQuant (Version 2.0.3.0) was used to annotate data. All data files were then analyzed in Skyline-daily (Version 22.2.1.351) to obtain peak areas for relative quantification of peptide abundance. Peaks were extracted for each target peptide under consideration of retention times. The mass spectrometry data have been deposited to the ProteomeXchange consortium via the jPOST repository with the dataset identifier PXD041784 for ProteomeXchange and JPST002140 for jPOST[60]. For data analysis, low-abundance peptides with peak areas below 10,000 mAU were excluded from the data set. Further, all peptides included in the analysis were identified at least twice among three biological replicates. The relative abundance of percent fragmentation at specific residues was normalized to the sum of the total peptide area identified for each sample.

### Primary neuron dissection and culturing

Primary hippocampal neurons were obtained from E18 CD-1 mice. Hippocampi were dissected in Hanks' Balanced Salts with 10 mM HEPES and penicillin/streptomycin, followed by digestion with 0.25% Trypsin-EDTA / 0.02 mg/mL DNase at 37 °C for 15 min and mechanical dissociation by trituration through a fire-polished Pasteur pipette. Neurons were then resuspended in plating medium (MEM supplemented with glucose, L-glutamine, 10% heat-inactivated horse serum, and penicillin/streptomycin) at a density of 25k cells/cm² on poly-L-Lysine coated coverslips in 24-well plates for ICC or at 80k cells/cm² in 96-well plates for viability assays. The media was then changed to

neuronal maintenance medium (Neuro basal medium with L-glutamine and B27 supplement and penicillin/streptomycin) after 2–4 h. Neurons were then treated with α-syn inhibition reactions on DIV 18-21. Here, samples from inhibition reactions were taken at the 48 h time point, sonicated, and applied to the neurons (1 µg for ICC or 7.2 µg for viability assays). Neuronal viability was assessed by MTT assay after 1 day, while aggregation was assessed by immunocytochemistry as described above after 1 week. Images were acquired using a Leica Sp8 Single Photon Confocal microscope with a 10x HC PL Apo CS2 objective, image acquisition was performed via Leica LAS X software, and processed with ImageJ 2.0.0.

### Neuronal viability

Cell viability was assessed by MTT assay. Here, 24 h following addition of inhibition reactions (DIV 19-22), viability was assessed by MTT Cell Proliferation Assay (ATCC) according to the manufacturer's protocol. Absorbance readings were taken at 570 nm with a reference filter of 630 nm on a BioTek EPOCH2 microplate reader.

### Data analysis software

All statistical analysis was performed in GraphPad Prism Version 9.3.1.

### Reporting summary

Further information on research design is available in the Nature Portfolio Reporting Summary linked to this article.

## Data availability

The mass spectrometry data generated in this study have been deposited in the ProteomeXchange consortium via the JPOST repository under the accession code: PXD044806 (HTRA1 α-Syn native-PAGE experiment) and PXD041784 (HTRA1 disaggregation and proteolysis experiment) for ProteomeXchange and JPST002295 and JPST002140 for jPOST. Source data are provided with this paper.

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

## Acknowledgements

We thank members of the Jackrel Lab for their feedback. We thank Peter Lansbury for α-synuclein plasmids and Tritia Yamasaki for the FRET biosensor cell lines. Our studies were supported by a Milton Safenowitz Postdoctoral Fellowship from the ALS Association (to A.P.), NIH grant F31NS120512 (to M.L.S.), NIH grant F31GM140622 (to J.J.R.), NIH grant K08NS101118 (to A.A.D.), and a grant from the Longer Life Foundation and NIH grant R35GM128772 (to M.E.J.).

## Author contributions

S.C. and A.P. designed and performed the experiments, analyzed data, and drafted the figures and portions of the manuscript. B.B., J.F., H.H.P., M.B., M.L.S., and M.K.H. performed experiments and analyzed data. J.J.R. and J.N.H. contributed key reagents and materials. G.J.P. supervised mass spectrometry experiments and A.A.D. assisted with the design, analysis, and supervision of mammalian cell experiments. M.E.J. designed and supervised the overall study, analyzed data, and wrote the manuscript. All authors reviewed and edited the manuscript.

## Competing interests

G.J.P. has a collaborative research agreement with ThermoFisher Scientific and is the Chief Scientific Officer of Panome Bio. All other authors declare no competing interests.
