## [Peer Review File · Nature Communications]

Reviewers' Comments:

Reviewer #1:

Remarks to the Author:

This work reports that HtrA1 can act against alpha-synuclein aggregation. This would be an important conclusion. However, the evidence provided is not sufficient to support it.

In Figure 2, the authors interpret the results in terms of alpha-synuclein and HtrA1 forming a stable complex that preserves the solubility of both proteins. This conclusion would seem in contradiction with the results in Figure 1, where HtrA1 degrades alpha-synuclein. Another possible explanation is that the inhibition of alpha-synuclein aggregation may be caused by the degradation of alpha-synuclein monomers. In any case, the ThT fluorescence should be shown as a function of time in the alpha-synuclein in vitro aggregation assay.

In Figure 3, the authors argue that HtrA1 makes PFFs seeding-incompetent. However, if the main effect of HtrA1 is to degrade alpha-synuclein monomers, the PFFs used in Figure 3 would not be formed in the presence of HtrA1. The authors should provide evidence that they have indeed treated cells with PFFs.

The complex formation reported in Figures 4 and S4 should be confirmed by additional experiments. In addition, the mass spec data should be shown.

It should be clarified whether or not the disaggregation activity shown in Figures 5 and 6 is due to the protease activity of HtrA1 on alpha-synuclein in the aggregated form. At the moment the evidence provided is not conclusive. This is essential, since the main conclusion of this work is that the disaggregase activity of HtrA1 is independent of its proteolytic activity.

The model that the authors propose in the Discussion for the disaggregase activity of HtrA1 (through the binding of the NAC region in the aggregates) should be illustrated by a figure, and the evidence for this model should be summarized more clearly. At the moment, it is not clear whether the evidence provided is consistent also with a protease-dependent disaggregate activity.

Contrary to what they write in the Introduction, the authors should recognize that there are approved drugs that clear amyloid aggregates, such as Aducanumab and Lecanemab.

Reviewer #2:

Remarks to the Author:

This manuscript provides biochemical and cellular evidence that the protease domain of HTRA1 inhibits alpha-synuclein aggregation and can protect cells. The data mostly supports the results. Most of the data is of good quality and supports the claims. I only have a few comments to address.

In figure 2C, there do not seem to be any bands for HTRA2 and it looks like HTRA2 promotes pelleting of alpha-syn. Is that real and why isn't this relected in 2D.

Is Fig. 7B all cells or only cells with HTRA1 expression? Can you clarify and show the data only for cells with HTRA1?

Why are you not seeing a dose response based on the expression in fig. 7C?

Is HTRA1 secreted? If so why are you seeing a signal in cells?

Reviewer #3:

Remarks to the Author:

In the manuscript entitled, "HtrA1 prevents and reverses alpha-synuclein aggregation, rendering it non-toxic and seeding incompetent", the authors study the ability of the protease family members

HTRA1 and HTRA2 to regulate α -synuclein (α -syn) misfolding, aggregation and seeding using both recombinant and cell culture systems. Focusing on HTRA1, they intriguingly find that the protease domain, but not necessarily protease activity is needed to prevent α -syn aggregation. Identification of molecules capable of resolving aggregates are appealing therapeutic targets and this manuscript provides mechanistic details on HTRA1's disaggregation and proteolysis of α -syn. However, there are some concerns about the specificity of HTRA1 proteolytic activity and its suitability as a therapeutic target that if addressed will be an exciting manuscript.

Major comments

The central finding of the paper is that HTRA1 disaggregase and proteolytic activity are two distinct mechanisms that can regulate α -syn aggregation and the formation of fibrils. If HTRA1's substrate is aggregated/fibril protein then both functions are needed, however it is not so clear how HTRA1 proteolytic activity is regulated.

- Can the authors comment/expand on the specificity of HTRA1 proteolysis. They show that HTRA1 does not cleave folded MBP or GST (Fig.1), but not other well-folded proteins. Also, in Fig 6 HTRA1 readily cleaves monomeric alpha synuclein. If proposed therapeutics increase HTRA1 levels, what proteins in addition to the intended target would be cleaved and would this be detrimental to the cellular viability? What are the known HTRA1 substrates?
- As they mention in the discussion (line 470), variants of HTRA1 could be used, perhaps the authors could be more explicit in their proposition of the proteolytic inactive HTRA1 as a possible therapeutic.

The authors find that HTRA1 can prevent the formation of amyloid fibrils for α -syn and inhibit aggregation of TDP-43 and FUS. They go on to solely focus on α -syn. Adding understanding of how HTRA1 prevents aggregation of FUS and TDP-43 would greatly enhance the importance of this manuscript. A single target that can be used to lower pathological burden across a range of diseases would be a very attractive therapeutic.

In Fig. 4, the authors examine the ability of the PDZ domain or the protease domain (ProD) wild-type and proteolytically deficient mutant to remodel α -syn.

- In some experiments full-length HTRA1 is used, while in other experiments different combinations of the other constructs are used. For the interpretation of these experiments, it would be important to include the entire panel of constructs for each experiment. Comparing the amyloid formation assays from Fig 4 to Fig 2, it seems like ProD alone is more efficient at preventing α -synuclein aggregation than the full-length construct, which would be interesting if proved true in the same experiment.
- In Fig 4B the authors perform a proteolysis assay using a FITC-casein substrate and then synuclein as the substrate in 4C. Full-length HTRA1 has increased proteolytic activity in panel 4B, whereas it appears the ProD alone has more activity in degrading α -synuclein. Could the authors comment on the differences, whether it is due to substrate specificity or an artifact from different assays?

In Figure 6, the authors show that pretreatment of α -syn PFFs with HTRA1 proteolytically dead mutant followed by addition of wild-type HTRA1 produced cleavage sites within the NAC domain of α -synuclein. These results suggest that HTRA1 disaggregates the fibrils allowing cleavage. Important controls would be to preincubate fibrils with the PDZ domain to reinforce that disaggregase activity is from the protease domain, and to treat PFF with wild-type HTRA1 followed by mutant to demonstrate without disaggregase activity, proteolysis of the NAC domain cannot happen.

In Fig. 5, the authors overexpress HTRA1 and proteolytic inactive mutant in cells and examine the seeding activity after adding α -syn PFF, finding that addition of HTRA1 WT or mutant decreases aggregation.

- In the Western blot associated with the figure the mutant is expressed several-fold more than the WT protein. Is this due to auto-proteolytic activity? If expressed at similar levels, would they see less of an effect of the mutant?
- In this cell line can endogenous levels of HTRA1 mitigate α -synuclein aggregation, for example compared to a knockdown?

- The authors state on line 365, "This activity is likely due to HtrA1 inhibiting α -synuclein aggregation and/or preventing the uptake of seeds." Does this suggest that disaggregase activity occurs extracellularly through secreted HTRA1? It would be important to know whether this process can happen within the cells or in the media or both for the development of future therapeutics.

In the α -syn fibril remodeling experiments Fig.2 and Fig. 5, the HTRA1 and protease-deficient HTRA1 display different ability to remodel fibrils which seems switched between experiments. In Fig. 2 addition of HTRA1 to α -syn prevents formation of fibrils leading to accumulation of amorphous material, whereas addition of the mutant led to more diffuse fibrils. This is somewhat opposite, when adding the HTRA1 protein to α -syn PFF. Is this due to different forms of α -syn, where the protease deficient HTRA1 works on the fibril material to disaggregate it?

The authors investigate the ability of HTRA1 to prevent amyloid formation and seeding activity of both wild-type and mutant α -syn in Fig. 2 and Fig. 3, respectively. Overall, they conclude, "... HtrA1 has native inhibitory and disaggregase activity against α -syn, but that this activity may be insufficient to overcome α -synA53T amyloidogenesis". While this conclusion is somewhat supported in the amyloid formation assay, Fig. 2A-B, this is not the case in the seeding assay, Fig. 3C-D, where it appears that HTRA1 prevents α -synuclein seeding competency similar for wild-type and mutant.

- It is difficult to compare the results regarding mutant and wild-type α -syn given the experiments were not performed head-to-head and presented in the same panel.
- Could the authors expand on these findings, is this result due to assays or substrate (α -syn in different forms) differences?
- Further, what domain of α -syn are critical for HTRA1 binding? Most of the cleavage occurs in the NAC domain (Fig. 4), but does binding happen in a distinct domain, like the amphipathic domain, where the A53T mutation resides? These studies would be interesting to explore.

In the discussion (line 421) the authors state, "...we anticipated that HtrA2 might be the principal HtrA isoform that mediates α -syn disaggregation." Further, they mention the link between HtrA2 mutations in PD. HtrA2 is primarily expressed in the mitochondrial inner membrane space and α -syn has been reported within various mitochondrial compartments, so it is plausible that HtrA2 could degrade mitochondrial α -syn. Could the authors expand on the rationale for investigating HTRA2 and its relevance to cytoplasmic α -syn?

Minor comments

As to avoid confusion, could the authors use HTRA1 for the human protein instead of HtrA1, which would be the nomenclature for the mouse protein?

In Fig 4B could the authors include the labels of constructs in the figure as well as the legend?

Could the authors include a section in the methods outlining what software they used to perform statistics.

In Figure 5F line 860, the authors state that N=3, however it appears there are 6 replicates in that panel.

In methods HtrA1 Δ NTD is mentioned as well as the protease-inactive mutant, but they don't appear in manuscript. If this construct is not used, could the authors remove it from the methods section.

Dear Reviewers,

Thank you for the review of our manuscript, NCOMMS-23-06087, *HtrA1 prevents and reverses α -synuclein aggregation, rendering it non-toxic and seeding incompetent*. We greatly appreciate the supportive reception of our work and all of the constructive feedback. We were happy to see that all three reviewers viewed our manuscript positively, though important concerns were also raised. Some of these concerns appear to stem from miscommunications due to some of the labels we used in figures, which we have modified as described below. We are grateful for the opportunity to revise our manuscript and have incorporated the requested revisions. We have performed many additional experiments and analyses. We have thoroughly revised each figure, and added a number of new panels and supplementary figures. We also thoroughly revised our text in response to the comments. Ultimately, our most significant conclusions have not changed with these newly added experiments, though we do believe that we have significantly strengthened the data supporting our conclusions. These new assays and analyses have also allowed us to add nuance to our model and proposed mechanism. For instance, we now show that silencing endogenous HTRA1 makes cells more vulnerable to α -syn seeding, suggesting that HTRA1 may have a native role in protecting against α -syn aggregation and toxicity. We also delineate the contributions of intracellular vs. secreted HTRA1 and find that the protective effects we observe are due to intracellular activity against α -syn rather than remodeling of extracellular seeds. We have also quantified dissociation constants to better understand the binding of HTRA constructs to α -syn and better define the mechanism of proteolytic-independent remodeling. We have also added a considerable amount of data characterizing the effects of HtrA1 against TDP-43 and FUS as requested. To aid in review, we have provided a point-by-point response to the concerns raised. Our comments are in bold. Edits to the manuscript are indicated with **yellow highlighting**.

Reviewer #1:

This work reports that HtrA1 can act against alpha-synuclein aggregation. This would be an important conclusion. However, the evidence provided is not sufficient to support it.

>> We thank the reviewer for recognizing the potential impact of our findings. Based on the helpful suggestions of the reviewer, we have accumulated additional evidence to support our conclusions, and we believe that our conclusions are now well-supported, as we describe below.

In Figure 2, the authors interpret the results in terms of alpha-synuclein and HtrA1 forming a stable complex that preserves the solubility of both proteins. This conclusion would seem in contradiction with the results in Figure 1, where HtrA1 degrades alpha-synuclein. Another possible explanation is that the inhibition of alpha-synuclein aggregation may be caused by the degradation of alpha-synuclein monomers. In any case, the ThT fluorescence should be shown as a function of time in the alpha-synuclein in vitro aggregation assay.

>> We thank the reviewer for this feedback and raising these important concerns. We believe that some of these issues stem from the phrasing we used in explaining some our results, but our results in these two figures are not contradictory. To clarify, it is important to note that the experiments in Figure 1 show proteolysis with HTRA1WT protein, while those in Figure 2 show degradation-independent inhibition of aggregation. In Figure 2 we used a proteolytically inactive mutant (HTRA1:S328A) in which the catalytic serine is mutated, rendering this variant proteolytically inactive and allowing us to monitor

disaggregation without the confounding effects of α -synuclein degradation. In Figure 2, we observe that HTRA1:S328A inhibits α -synuclein aggregation and preserves its solubility, but there is no degradation noted for α -synuclein or HTRA1:S328A. Degradation of α -synuclein is only noted when HTRA1WT is added, matching what we observe in Figure 1 for the HTRA1WT protein. Thus, the results from these two figures and across these two variants cannot be directly compared, and we feel that our conclusions that α -synuclein and HTRA1 form a stable complex that preserves the solubility of both proteins are well-founded. This conclusion is further supported by our newly added fluorescence polarization assays, where we show that HTRA1:S328A binds α -synuclein with a K_d of ~ 31 nM, which is relatively tight for a protein—protein interaction (Figure 4I).

We agree with the reviewer that we cannot rule out that the inhibitory activity of HTRA1WT (but not the S328A mutant) could alternatively be explained by degradation of α -synuclein monomer. Because we have not identified a way to specifically ablate the aggregation inhibition activity of HTRA1 while retaining its proteolytic activity, we cannot rule out this possibility for HTRA1WT. It is for this reason that use of the HTRA1:S328A mutant is crucial. Nonetheless, our results do indicate that inhibition of α -synuclein aggregation by HTRA1:S328A can be achieved without degradation. Further, our native PAGE experiments (Figure S4A) show the formation of a stable complex with α -syn by not just HTRA1:S328A, but also HTRA1WT. Though some degradation in the HTRA1WT lane is noted, the higher molecular weight band suggests complex can form despite proteolysis. Thus, we now have three lines of evidence of complex formation: the pull down assay, native gel experiment, and fluorescence polarization. To clarify these points, we have also edited the results section describing Figure 2 to more explicitly describe these key ideas. We have also edited our phrasing throughout the manuscript to clarify when certain statements may only apply to HTRA1:S328A.

Finally, we agree with the reviewer that it is important to show the ThT fluorescence assays as a function of time in the α -synuclein in vitro aggregation assay. We have acquired new data to show both α -synuclein WT and α -synuclein:A53T aggregation over time. This data is now shown in Figure 2A-B. As the bar graphs in the original manuscript would now be redundant with these time course assays, we have removed the original versions of these two panels.

In Figure 3, the authors argue that HtrA1 makes PFFs seeding-incompetent. However, if the main effect of HtrA1 is to degrade α -synuclein monomers, the PFFs used in Figure 3 would not be formed in the presence of HtrA1. The authors should provide evidence that they have indeed treated cells with PFFs.

>> We appreciate the reviewer's concern, and we believe that some of these concerns are in part due to the same miscommunication described in the previous point, as well as the way in which we labeled the images in Figure 3. However, we agree with the reviewer that it is of great importance to establish that we have indeed treated the cells with PFFs. Thus, we have also performed additional experiments to support these conclusions. To clarify, as described in the response to the point above, for the experiments in Figure 3, we are forming our " α -synuclein PFFs" in the presence of HTRA1, and we find that when HTRA1 is present these "PFFs" are either degraded prior to or in early stages of fibrillization (when treated with HTRA1WT) or remodeled. They do not actually fibrillize. The purpose of the experiments shown in the initial manuscript version was to show that regardless of what specific species forms (i.e. degraded, remodeled, or solubilized α -synuclein) that these HTRA-treated products would be incapable of seeding α -synuclein aggregation. We now

recognize that our labels in Figure 3 were misleading and we should not have used the term “PFFs” here because HTRA1 prevents fibrillization, as the reviewer points out. True PFFs are only formed for the α -synuclein + buffer condition in this figure. We have relabeled the images in in Figure 3B (now 3C) and edited the text to clarify these points.

We also recognize the importance of the second half of the reviewer’s comment, that we should provide evidence that α -synuclein PFFs form that are capable of entering cells, and that we are indeed treating the cells with PFFs. To address this, we have generated Alexa-568 labeled α -synuclein and repeated the experiment, incubating with buffer, HTRA1WT, or HTRA1:S328A. The label allows us to monitor α -synuclein internalization and aggregation by flow cytometry. Our results indicate that HTRA1 or HTRA1:S328A treatment does not have any significant effect on α -synuclein internalization (see new Figure 3B). We also confirmed that we observe the same remodeling by HTRA1WT and HTRA1:S328A when we use unlabeled or Alexa-568 labeled α -syn (see new Figure S3A-C). We have also added clarifying statements throughout the results section to aid interpretation.

We also point out that in Figure 5 we show disaggregation, where we form PFFs, validate PFFs are formed by ThT and EM, and subsequently treat the PFFs with HTRA1WT or HTRA1:S328A. Here we again see that this HTRA1 treatment renders the remodeled products seeding-incompetent.

The complex formation reported in Figures 4 and S4 should be confirmed by additional experiments. In addition, the mass spec data should be shown.

>> To confirm that complex formation is occurring, we have now quantified the affinity of this interaction using fluorescence polarization assays. Here, we incubated α -synuclein-Cys-Alexa 488 with HTRA1:S328A, HTRA1ProD:S328A, or HTRA1PDZ. Binding to HTRA1:S328A and HTRA1ProD:S328A is relatively tight, with a K_d of ~ 31 nM and ~ 74 nM, respectively. This data is now shown in Figure 4I.

As requested, we have added the in-gel digest mass spectrometry data of the native-PAGE shown in Fig S4. This data further supports complex formation, indicating that α -synuclein and HTRA1 were found together in a single band on a gel. Therefore, we now have three distinct lines of evidence of complex formation: the pull down assay, native gel experiment, and fluorescence polarization.

It should be clarified whether or not the disaggregation activity shown in Figures 5 and 6 is due to the protease activity of HtrA1 on alpha-synuclein in the aggregated form. At the moment the evidence provided is not conclusive. This is essential, since the main conclusion of this work is that the disaggregase activity of HtrA1 is independent of its proteolytic activity.

>> We believe that this comment is largely due to the miscommunication about the distinction between HtrA1WT and HTRA1:S328A discussed above. In Figures 5 and 6 we again use the proteolytically inactive HTRA1:S328A variant to tease apart the contributions of proteolytic and disaggregase activity. We find that, like the inhibitory activity, the disaggregase activity of HTRA1 is also independent of proteolysis. In Figure 5, we employ both HTRA1WT and the proteolytically inactive HTRA1:S328A variant. We find that both proteins are capable of disaggregating α -synuclein PFFs, indicating that proteolysis is not required for disaggregation. We have further clarified these points in the results section,

and throughout the manuscript we now more explicitly state which comments relate to just HTRA1WT or HTRA1:S328A.

We also agree with the reviewer that, based on this data, we cannot exclude the possibility that the protease activity of HTRA1WT may also contribute to disaggregation. To further investigate this idea, we have now compared HTRA1 WT-mediated proteolysis of α -syn monomer to PFFs (Fig S6C). We observe nearly complete proteolysis of monomeric α -syn and minimal proteolysis of the fiber. Further, HTRA1:S328A is nearly as active as HTRA1WT in disaggregation. Thus, while HTRA1WT disaggregase activity may rely in part on proteolysis, HTRA1:S328A disaggregase activity is independent of proteolysis. HTRA1:S328A disaggregase activity is rather similar to that of the WT protein, suggesting that proteolysis of α -syn fibrils is not a significant contributor to the disaggregation of α -syn PFFs that we observe. Thus, we believe that our conclusions that disaggregation primarily occurs on the fibrillar form without requiring protease activity is well-supported. Our results in Figure 6 are consistent with all of these points, and we have also added additional controls to Figure 6 (described below) to further strengthen this data.

The model that the authors propose in the Discussion for the disaggregase activity of HtrA1 (through the binding of the NAC region in the aggregates) should be illustrated by a figure, and the evidence for this model should be summarized more clearly. At the moment, it is not clear whether the evidence provided is consistent also with a protease-dependent disaggregate activity.

>> We thank the reviewer for this feedback and in the revised manuscript we have tried to summarize this evidence more clearly. Further, with our newly added experiments for this revision, we believe that the support for this model is now more convincing. As suggested, we have developed an illustration to explain our model, which is now shown in Figure 7. To summarize, we propose that HTRA1 can both disaggregate and proteolyze α -synuclein aggregates, and these two processes can be considered distinct functions of the protein. We can differentiate these two processes by use of HTRA1:S328A, which can disaggregate fibrils but not proteolyze them. In this figure, we show that HTRA1WT can ordinarily cleave α -synuclein monomer throughout the α -synuclein sequence. Once α -synuclein forms fibrils (second row), HTRA1 can still proteolyze α -synuclein, however these digestion sites are restricted to the N- and C-terminal regions, which are more accessible. The NAC domain remains protected from cleavage. If HTRA1:S328A is added as a pretreatment step, allowing disaggregation but not proteolysis, we note that the NAC domain is no longer protected, and once HTRA1WT is added proteolysis can now occur in the NAC domain. The primary evidence for this model comes from the data shown in Figure 6, where we modulated the order of addition of HTRA variants followed by proteolysis to identify cleavage products. In this revised version, we add two key controls to this experiment, which further support our conclusions. Importantly, if just HTRA1WT is added, proteolysis of the NAC domain does not occur, indicating that our data is consistent with a protease-independent disaggregation phenomenon. In addition to adding these experiments and the new summary figure, we have edited the text in both the results and discussion sections to better describe this evidence and our model.

Contrary to what they write in the Introduction, the authors should recognize that there are approved drugs that clear amyloid aggregates, such as Aducanumab and Lecanemab.

>> We thank the reviewer for pointing this out. We have now revised the introduction and added references and discussion of recent advancements in drug development to clear amyloid aggregates in AD (Sevigny, Chiao et al. 2016, van Dyck, Swanson et al. 2022, Reardon 2023). Indeed, though it remains controversial if these therapeutics modulate disease progression in a sufficiently meaningful way to merit widespread use, they provide further support of the relevance of targeting HTRA1 as a therapeutic. Advances with Aducanumab and Lecanemab provide proof-of-concept that clearance of amyloid aggregates can be an effective strategy in patient populations (see lines 67-72).

Reviewer #2:

This manuscript provides biochemical and cellular evidence that the protease domain of HTRA1 inhibits α -synuclein aggregation and can protect cells. The data mostly supports the results. Most of the data of good quality and supports the claims. I only have a few comments to address.

>> We appreciate the reviewer's supportive feedback. We have now addressed each of the reviewer's questions in the revised manuscript as described below.

In figure 2C, there do not seem to be any bands for HTRA2 and it looks like HTRA2 promotes pelleting of α -syn. Is that real and why isn't this reflected in 2D.

>> We thank the reviewer for pointing this out, and our data suggests that HTRA2 does not inhibit pelleting of α -synuclein into insoluble aggregates, though it does not necessarily *promote* pelleting. Looking back through our sedimentation assay trials, we do see variability from trial to trial with respect to the amount of α -synuclein in the pellet and soluble fractions upon incubation with HTRA2. We believe that this variability is due to varying kinetics of autoproteolysis of HTRA2, which we find varies among batches of purified protein. It is this autoproteolysis that is responsible for the missing bands for HTRA2. Nonetheless, all four trials show that only a small amount of α -synuclein remains soluble following HTRA2 treatment. However, we now recognize that we did not select the best trial to be truly representative of the four trials and best reflect the graphical representation in Figure 2D. We have now replaced the representative gel shown in Figure 2C with a different trial that is more representative, particularly with respect to the HTRA2 data. We have also included the gels for all four trials in the Source Data file.

Is Fig. 7B all cells or only cells with HTRA1 expression? Can you clarify and show the data only for cells with HTRA1?

>> We thank the reviewer for pointing out this key question. In the original manuscript, the data in Figure 7B (now 8B) does show α -synuclein aggregation for all cells. We initially did not perform immunolabeling against HTRA1 for the flow cytometry experiment due to technical challenges, such as antibody specificity and variable labeling efficiency.

We agree with the reviewer that it would be useful to analyze the data showing only cells expressing HTRA1. As requested, we have now performed additional troubleshooting to improve our immunolabeling against HTRA1 using an anti-Myc antibody and quantified the Myc-positive cells overexpressing HTRA1 by flow cytometry. We observed that only ~5% of the cell population was Myc-positive upon induction of HTRA1WT and ~15% of cells were Myc-positive in HTRA1:S328A cells (Fig S8F). The lower percentage of Myc-positive cells in the HTRA1WT population is presumably due to autoproteolysis of HTRA1, thereby decreasing the number of Myc-positive cells.

We then gated to select Myc-positive cells for quantification of α -synuclein seeding for only the population of cells expressing HTRA1. We find an ~50% reduction in α -synuclein aggregation upon HTRA1WT overexpression and an ~30% reduction in α -synuclein aggregation upon HTRA1:S328A overexpression as compared to vector control. These results are consistent with our original data shown in Figure 7B, demonstrating that HTRA1 overexpression can prevent α -synuclein seeding in the cellular environment.

However, these new experiments are somewhat limited, as we are concerned that our labeling is still suboptimal. While our results are consistent with the data presented in Figure 7B, we are concerned by these issues and have decided not to repeat the dose-dependence experiments in this format, but to instead include the new experiment as a supplementary figure (Fig S8F-I). We also clarify that the data shown in Figure 8B is for all cells. Further, in response to a request from Reviewer 3, we have silenced the HTRA1 gene via siRNA and then performed similar assays. These results show that silencing the HTRA1 gene increases the vulnerability of the cells to seeding by α -synuclein, providing further evidence that these effects are HTRA1-specific (Figure 8H-I).

Why are you not seeing a dose response based on the expression in Fig. 7C?

>> We believe that the reviewer is asking why we do not see a stronger inhibition of seeding given the apparently much higher levels of expression of HTRA1:S328A for the “high” population as compared to the “low” population. First, it is important to note that for both HTRA1WT and HTRA1:S328A, at higher HTRA1 expression levels we do observe greater inhibition of α -synuclein seeding, consistent with a dose response. Nonetheless, we suspect that the reviewer is asking why we are not seeing a larger magnitude of change that follows the expression levels from the immunoblot. We believe that there are several factors responsible for this. First, the immunoblots for HTRA1WT levels are somewhat misleading due to autoproteolysis of HTRA1WT. Thus, no dose dependent changes are noted on the immunoblot, yet we do see a dose-dependent decrease in seeding. Also of note, we observe that the bands corresponding to degradation products are more intense for the higher HTRA1 expression level samples. We do not observe autoproteolysis for HTRA1:S328A because it is proteolytically inactive, and so we see a dose dependent change in expression level here. As with WT, we also see a dose dependent decrease in seeding for these samples. Because we observe greater inhibition at higher expression levels for both HTRA1WT and HTRA1:S328A, we believe that the inhibitory effects are most effective at early stages, before autoproteolysis occurs. To explore this further, we acquired additional immunoblots 24h and 48h post-transfection (Figure S8B). At 24hr, we observe an increase in HTRA1 expression corresponding with higher transfection for both the HTRA1WT and HTRA1:S328A plasmids. At 48h, the autoproteolysis of HTRA1 is much greater than at 24h, and so we no longer observe the increased expression, though we again note that the bands for the degradation products are more intense for the samples transfected with more plasmid.

It is unclear why we do not see a larger magnitude of an effect in the FRET seeding assay, however, we suspect that perhaps there is a maximum protective effect that HTRA1 can achieve, and even at these higher levels of HTRA1 expression, the cells cannot fully overcome the proteotoxic seeding of the α -synuclein PFFs.

Is HTRA1 secreted? If so why are you seeing a signal in cells?

>> Htra1 contains an N-terminal signal sequence responsible for its secretion, but it has also been shown to be retained inside the cell (Globus, Evron et al. 2017). In the immunoblots we showed in the original manuscript, only the cell lysates were run, and so the observed signal was just due to intracellular HTRA1.

However, the reviewer raises a very interesting point, which was also raised by Reviewer 3. Therefore, we decided to investigate these ideas further. To do so, we designed a new

NTD deletion construct to prevent HTRA1 secretion. As shown in Figure 8D-G, we observe decreased secretion of this HTRA1:ΔNTD construct in the media. We then repeated the FRET seeding assay and we observe that deletion of the NTD leads to a significant increase in seeding inhibition, indicating that HTRA1 is primarily countering seeding by α-synuclein once it enters the cell, rather than in the media.

Reviewer #3 (Remarks to the Author):

In the manuscript entitled, "HtrA1 prevents and reverses α -synuclein aggregation, rendering it non-toxic and seeding incompetent", the authors study the ability of the protease family members HTRA1 and HTRA2 to regulate α -synuclein (α -syn) misfolding, aggregation and seeding using both recombinant and cell culture systems. Focusing on HTRA1, they intriguingly find that the protease domain, but not necessarily protease activity is needed to prevent α -syn aggregation. Identification of molecules capable of resolving aggregates are appealing therapeutic targets and this manuscript provides mechanistic details on HTRA1's disaggregation and proteolysis of α -syn. However, there are some concerns about the specificity of HTRA1 proteolytic activity and its suitability as a therapeutic target that if addressed will be an exciting manuscript.

>> We appreciate the reviewer's positive feedback and recognition of the potential impact of our study. We believe that we have been able to effectively address each of the reviewer concerns, and our responses are described below.

Major comments

The central finding of the paper is that HTRA1 disaggregase and proteolytic activity are two distinct mechanisms that can regulate α -syn aggregation and the formation of fibrils. If HTRA1's substrate is aggregated/fibril protein then both functions are needed, however it is not so clear how HTRA1 proteolytic activity is regulated.

>> We appreciate the reviewer's suggestion, and regulation of HTRA1 proteolytic activity has been heavily explored by others. We have now added further discussion of this regulation to the revised text in the introduction (lines 92-94). We have also added two citations that discuss this regulation further (Cabrera, Melo et al. 2017, Rey, Breiden et al. 2022).

• Can the authors comment/expand on the specificity of HTRA1 proteolysis. They show that HTRA1 does not cleave folded MBP or GST (Fig.1), but not other well-folded proteins. Also, in Fig 6 HTRA1 readily cleaves monomeric alpha synuclein. If proposed therapeutics increase HTRA1 levels, what proteins in addition to the intended target would be cleaved and would this be detrimental to the cellular viability? What are the known HTRA1 substrates?

>> We thank the reviewer for pointing this out. Our data suggests that HTRA1 selectively cleaves unfolded or misfolded proteins, but does not have this broad activity and is more selective against well-folded proteins. In Figure 1D-E we observe that HTRA1 robustly cleaves α -synuclein monomer, which is intrinsically disordered. Similarly, after separation of MBP-TDP-43 and GST-FUS with TEV protease, we note strong cleavage of TDP-43 and FUS, which harbor prion-like domains and are highly aggregation-prone. However, we note no cleavage of MBP or GST, which are well-folded, even after 24h of incubation. To further address this question, we performed additional experiments in which we treated purified GST or casein (a soluble, yet unfolded, substrate) with HTRA1 and monitored proteolysis. We note no proteolysis of GST after a 24h treatment, while proteolysis of casein is complete in this timeframe. This new data is now shown in Figure S1A-B.

If new therapeutics were designed to increase HTRA1 levels, we would expect that the entire HTRA1 substrate repertoire would also be subject to increased proteolysis. We find that HTRA1 can proteolyze casein, TDP-43, FUS, and α -synuclein. Others have reported

that HTRA1 can proteolyze substrates including tau, A β , extracellular matrix proteins, elastin, fibronectin, and aggrecan. As the reviewer recognizes, increasing levels of such a protease this may not translate directly into a feasible therapeutic strategy. We anticipate that upregulating HTRA1 levels would not be as effective as introducing an HTRA1 construct that can disaggregate substrates yet lacks proteolytic activity (such as HTRA1ProD:S328A). This would allow for this new construct to dissolve aggregates yet preserve full-length HTRA1 at endogenous levels and not interfere with its native activity. To address these points, in addition to the new data with TDP-43, FUS, and casein, we have also added additional clarification of these ideas to the results section and expansion of our discussion of the therapeutic strategy that our results would support in the discussion section (lines 581-586).

• As they mention in the discussion (line 470), variants of HTRA1 could be used, perhaps the authors could be more explicit in their proposition of the proteolytic inactive HTRA1 as a possible therapeutic.

>> As we discuss above, this is a very important idea that merits further discussion. We have now revised the discussion section to more explicitly and comprehensively discuss our ideas for therapeutic translation (lines 581-586).

The authors find that HTRA1 can prevent the formation of amyloid fibrils for α -syn and inhibit aggregation of TDP-43 and FUS. They go on to solely focus on α -syn. Adding understanding of how HTRA1 prevents aggregation of FUS and TDP-43 would greatly enhance the importance of this manuscript. A single target that can be used to lower pathological burden across a range of diseases would be a very attractive therapeutic.

>> We thank the reviewer for this suggestion. We intended to focus on α -synuclein because we were concerned that adding a more significant amount of data related to TDP-43 and FUS might make the manuscript too broadly focused. However, we agree that the identification of a single target that can lower the pathological burden across a range of diseases would be very attractive.

As a result, we have expanded our study to add this new data. We now include data for TDP-43 and FUS for most of the in vitro experiments. This new data is shown in Supplementary Figures 1 and 5, along with discussion of these experiments in the results and discussion sections. To briefly summarize our findings (please see the revised manuscript for a full description of the new data), we now compare the activities of HTRA1 and HTRA1:S328A to demonstrate that activity against TDP-43 and FUS is also proteolytically-independent. We also probed the same series of HTRA1 constructs assayed for α -syn aggregation inhibition against TDP-43 and FUS in proteolysis and turbidity assays. Similarly to our findings with α -synuclein, we find that HTRA1 robustly proteolyzes FUS while HTRA1:S328A, HTRA1:ProDS328A, and HTRA1:PDZ do not proteolyze FUS. We also see that HTRA2 has weaker proteolysis activity against FUS as compared to HTRA1. When monitoring inhibition of aggregation via turbidity, we observe that HTRA1WT, HTRA1:S328A, HTRA1ProD, and HTRA1ProD:S328A all inhibit FUS aggregation while HTRA1:PDZ does not. Consistent with our studies with α -synuclein, we find that the protease domain is also sufficient for proteolysis of FUS and inhibition of FUS aggregation.

We observe similar results with TDP-43, again finding that HTRA1 can inhibit TDP-43 aggregation through a proteolytically-independent mechanism. Therefore, as the reviewer proposes, we can now conclude that HTRA1 might be a viable single target that can be used to lower the pathological burden across not just PD, but also ALS/FTD, making this an attractive therapeutic target.

In Fig. 4, the authors examine the ability to the PDZ domain or the protease domain (ProD) wild-type and proteolytically deficient mutant to remodel α -syn.

- In some experiments full-length HTRA1 is used, while in other experiments different combinations of the other constructs are used. For the interpretation of these experiments, it would be important to include the entire panel of constructs for each experiment. Comparing the amyloid formation assays from Fig 4 to Fig 2, it seems like ProD alone is more efficient at preventing α -synuclein aggregation than the full-length construct, which would be interesting if proved true in the same experiment.

>> We appreciate the review's suggestion and agree that including the entire panels of constructs for each experiment would allow for direct comparison and improve our ability to interpret our results. As requested, we have now repeated the indicated experiments shown in Figure 4 including the full-length HTRA1 control alongside all of the other constructs, allowing for direct comparison. These new panels include Figure 4C, D, E, and F. Based on the prior comment, we also expanded these experiments to include TDP-43 and FUS (Figure S5). Overall, our conclusions have not changed, though we can now make direct comparisons across all of these constructs. We agree that it would be very interesting if HTRA1ProD alone was more efficient than full-length HTRA1 in preventing α -synuclein aggregation. However, we find that there was no statistical difference in the activity levels of these two proteins. Thus, we can conclude that the inhibitory activity of HTRA1 against α -synuclein, TDP-43, and FUS requires the protease domain and that the PDZ domain is dispensable for this activity.

- In Fig 4B the authors perform a proteolysis assay using a FITC-casein substrate and then synuclein as the substrate in 4C. Full-length HTRA1 has increased proteolytic activity in panel 4B, whereas it appears the ProD alone has more activity in degrading α -synuclein. Could the authors comment on the differences, whether it is due to substrate specificity or an artifact from different assays?

>> We thank the reviewer for pointing this out. First, we do not feel that these two experiments should be directly compared given the different substrates tested, and that panel 4B is performed over 80 minutes and panel 4C over 24h. In panel 4B, we can say that HTRA1 has increased proteolytic activity against casein as compared to HTRA1ProD. In panel 4C, we actually see complete degradation of α -synuclein by both HTRA1 and HTRA1ProD. We believe that this confusion is due to a band that appears in the HTRA1 lane that runs at a slightly higher MW than undigested α -synuclein (compare the bands in the buffer lane to HTRA1 lane). To confirm this, we have added an HTRA1 alone control. Here we see the same band is generated, despite no addition of α -synuclein, confirming that this band is a product of HTRA1 autoproteolysis.

To directly compare the activities of HTRA1 and HTRA1ProD, we have performed an additional proteolysis assay where we monitored the time course of proteolysis. This new

data is shown in Figure S4C. We find that the results are fairly similar between the full length and HTRA1ProD constructs, with digestion of α -syn nearly complete by 8h. This suggests that the PDZ domain does not modulate the binding of HTRA1 to α -syn.

In Figure 6, the authors show that pre-treatment of α -syn PFFs with HTRA1 proteolytically dead mutant followed by addition of wild-type HTRA1 produced cleavage sites within the NAC domain of α -synuclein. These results suggest that HTRA1 disaggregates the fibrils allowing cleavage. Important controls would be to preincubate fibrils with the PDZ domain to reinforce that disaggregase activity is from the protease domain, and to treat PFF with wild-type HTRA1 followed by mutant to demonstrate without disaggregase activity, proteolysis of the NAC domain cannot happen.

>> We agree with the reviewer that these are important controls. As requested, we have preincubated the fibrils with the PDZ domain followed by HTRA1WT, and find that the NAC domain remains cleavage-resistant as expected. This data is shown in Supplementary Figure 7. While it would be interesting to pretreat with HTRA1 WT followed by mutant, we were concerned that proteolysis of the NAC domain might still occur after the addition of HTRA1SA mutant due to the disaggregation activity of the mutant while WT remains present. Instead, we decided to preincubate with HTRA1ProDSA followed by HTRA1WT and added this experiment to the data shown in Figure 6B-D. We find that pretreatment with ProDSA results in a cleavage pattern and number of cleavage sites highly similar to that of HTRA1:S328A pretreatment. Thus, both SA and ProDSA allow HTRA1WT to proteolyze the PFFs at the NAC domain.

Further, we find that PDZ domain pretreatment resulted in a substantial decrease in fragmentation, and cleavage was largely localized to the C-terminal region of α -synuclein rather than in the NAC domain. Taken together, these new controls support our conclusions that the protease domain of HTRA1 is responsible for disaggregation of α -syn PFFs.

In Fig. 5, the authors overexpress HTRA1 and proteolytic inactive mutant in cells and examine the seeding activity after adding α -syn PFF, finding that addition of HTRA1 WT or mutant decreases aggregation.

- In the Western blot associated with the figure the mutant is expressed several-fold more than the WT protein. Is this due to auto-proteolytic activity? If expressed at similar levels, would they see less of an effect of the mutant?

>> We believe that there is a typo here and the Reviewer is referring to Figure 7 (now Figure 8). As the reviewer points out, we observe much higher levels of HTRA1:S328A as compared to HTRA1WT. This is due to auto-proteolysis of HTRA1WT. This is reflected in the increased intensity of the lower MW bands for the “HTRA1WT (high)” lane, which correspond to degradation products. We repeated these immunoblots with samples collected 24 and 48h post-transfection and find that the expression levels are more comparable between the two constructs (Supplementary Figure S8B).

- In this cell line can endogenous levels of HTRA1 mitigate α -synuclein aggregation, for example compared to a knockdown?

>> This is an excellent suggestion. To investigate if endogenous levels of HTRA1 can mitigate α -synuclein aggregation, we followed the reviewer's suggestion and performed HTRA1 siRNA knockdown in the HEK293T- α -syn biosensor cells. We validated knockdown by RT-qPCR and observe greater than a 20-fold decrease in HTRA1 mRNA levels. We then seeded with α -syn PFFs to examine the activity of endogenous HTRA1. These new results are shown in Figure 8H-I. We note that knockdown of HTRA1 increases α -synuclein aggregation by ~40%, indicating that endogenous HTRA1 can mitigate the effects of α -synuclein PFFs.

• The authors state on line 365, "This activity is likely due to HtrA1 inhibiting α -synuclein aggregation and/or preventing the uptake of seeds." Does this suggest that disaggregase activity occurs extracellularly through secreted HTRA1? It would be important to know whether this process can happen within the cells or in the media or both for the development of future therapeutics.

>> This is an excellent suggestion, and a similar suggestion was made by Reviewer 2. We have now expanded on this idea with new experiments. We hypothesized that by removing the NTD secretion signal from HTRA1, the protein would be retained in the cells. We therefore designed an HTRA1 Δ NTD construct and repeated these assays. As shown in Figure 8D-G, we observe decreased secretion of this HTRA1: Δ NTD construct in the media. We then repeated the FRET seeding assay and we observe that deletion of the NTD leads to a significant increase in seeding inhibition, indicating that HTRA1 is primarily countering seeding by α -synuclein once it enters the cell, rather than in the media. To further validate this idea, we performed α -syn cellular internalization experiments using Alexa-568 labeled α -syn PFFs. We find that α -syn PFFs are taken up similarly in cells expressing HTRA1WT and HTRA1: Δ NTD (Figure 8D). These new experiments refine our understanding of the HTRA1 mechanism, and so we have modified the text accordingly (lines 438-447) and in the discussion.

In the α -syn fibril remodeling experiments Fig. 2 and Fig. 5, the HTRA1 and protease-deficient HTRA1 display different ability to remodel fibrils which seems switched between experiments. In Fig. 2 addition of HTRA1 to α -syn prevents formation of fibrils leading to accumulation of amorphous material, whereas addition of the mutant led to more diffuse fibrils. This is somewhat opposite, when adding the HTRA1 protein to α -syn PFF. Is this due to different forms of α -syn, where the protease deficient HTRA1 works on the fibril material to disaggregate it?

>> We thank the reviewer for pointing this out, and now see how these results might appear to have been switched. However, we can confirm from multiple trials that there was no mixup of the samples. While we did our best to find representative images to show in the manuscript, TEM images are qualitative by nature, hence the need to perform quantitative assays alongside. There were also several important experimental differences in these assays that may be responsible for these differences. In Figure 2, we used a 5:1 molar ratio of α -synuclein to HTRA1, while in Figure 5 we used a 1:10 ratio with excess HTRA1 to drive disaggregation.

We found that fibril remodeling requires a much higher molar concentration of HTRA1 as compared to inhibition. With this higher amount of HTRA1 in Figure 5, we achieved a greater overall decrease in α -syn fibrils, and so we do not think these two experiments should be compared directly. To address this issue, we have now added several views of

each experimental condition to Supplementary Figure 2D and Supplementary Figure 6A. These additional panels are more representative, showing that HTRA1 WT nearly completely inhibits fibrillization while fibrils are more abundant in the HTRA1:S328A images, though all appear less tightly wound than the untreated fibrils. Similar effects are observed for the disaggregation assay samples. These morphological changes are supported by our ThT and cellular seeding assays, which show that this remodeling distinctly changes the properties of these materials. We have also edited the text to more thoroughly explain these observations.

The authors investigate the ability of HTRA1 to prevent amyloid formation and seeding activity of both wild-type and mutant α -syn in Fig. 2 and Fig. 3, respectively. Overall, they conclude, "... HtrA1 has native inhibitory and disaggregase activity against α -syn, but that this activity may be insufficient to overcome α -synA53T amyloidogenesis". While this conclusion is somewhat supported in the amyloid formation assay, Fig. 2A-B, this is not the case in the seeding assay, Fig. 3C-D, where it appears that HTRA1 prevents α -synuclein seeding competency similar for wild-type and mutant.

- It is difficult to compare the results regarding mutant and wild-type α -syn given the experiments were not performed head-to-head and presented in the same panel.

>> We thank the reviewer for pointing this out. Indeed, it is difficult to compare the results from wild-type α -synuclein and mutant α -synuclein directly since wild-type and mutant α -synuclein aggregate with different kinetics and morphologies. In our ThT assays, we note α -synuclein WT and A53T produce different total levels of ThT fluorescence, which is consistent with reports in the literature (Conway, Lee et al. 2000, Flagmeier, Meisl et al. 2016). Thus, we agree with the reviewer and we do not think that the results from with the WT and A53T mutant should be directly compared. To address this point, we have revised the manuscript to describe the inhibitory activity of HTRA1 against α -synuclein:WT and A53T without directly comparing these assays.

- Could the authors expand on these findings, is this result due to assays or substrate (α -syn in different forms) differences?

>> As we mention above, we have decided it is best not to compare these results directly. Nonetheless, we believe that likely these differences are due both to the assays and due to the differences in the two substrates. First, the two assays report on different features. In the ThT assay, we are monitoring total amyloid content. However, it is plausible that some materials may be ThT-reactive and unable to seed α -synuclein aggregation, and vice versa. Further, we do believe that there are inherent differences between the WT and A53T substrates. Similar questions were also raised by Reviewer 1. Therefore, we decided to investigate further by repeating the ThT amyloid formation assays as a function of time. The data is now shown in Fig 2A-B.

- Further, what domain of α -syn are critical for HTRA1 binding? Most of the cleavage occurs in the NAC domain (Fig. 4), but does binding happen in a distinct domain, like the amphipathic domain, where the A53T mutation resides? These studies would be interesting to explore.

>> We appreciate the reviewer's suggestion and this would be an interesting idea to further explore. We had tried to explore this by constructing different α -syn fragments and probing binding by fluorescence anisotropy. Unfortunately, we found that these peptides produced very high background levels of anisotropy, precluding measurement of binding affinity. We believe that this is due to aggregation of the NAC domain. As developing new peptides and screening to ensure solubility would be a large undertaking, we suggest that these studies be avenues of future exploration.

In the discussion (line 421) the authors state, "...we anticipated that HtrA2 might be the principal HtrA isoform that mediates α -syn disaggregation." Further, they mention the link between HtrA2 mutations in PD. HtrA2 is primarily expressed in the mitochondrial inner membrane space and α -syn has been reported within various mitochondrial compartments, so it is plausible that HtrA2 could degrade mitochondrial α -syn. Could the authors expand on the rationale for investigating HTRA2 and its relevance to cytoplasmic α -syn?

>> We thank the reviewer for the pointing this out. It was previously reported that α -synuclein accumulates in the mitochondria in Parkinson's disease patient brains, leading to reduced mitochondrial activity in Parkinson's disease. Thus, we were interested in investigating if HTRA2 could protect against α -synuclein aggregation. We ultimately decided it would be best to directly compare HTRA1 and HTRA2, and so we restricted our assays with HTRA2 primarily to biochemistry assays to avoid the caveat of subcellular localization. Nonetheless, because α -syn seeds would first be internalized into the cytoplasm, we believed it would be important to investigate the role of HTRA2 against these materials and so we performed some cellular assays with HTRA2. It would be interesting to further test the effects of HTRA2 in greater detail, including cell biological experiments to query organelle-specific functions of HTRA2. We suggest that such studies would best be the subject of future work. We have expanded the indicated excerpt in the second paragraph of the discussion to include rationale for investigating the relevance of HTRA2 to cytoplasmic α -syn.

Minor comments

As to avoid confusion, could the authors use HTRA1 for the human protein instead of HtrA1, which would be the nomenclature for the mouse protein?

>> We have revised the text and figures accordingly.

In Fig 4B could the authors include the labels of constructs in the figure as well as the legend?

>> We have added a legend to the panel of this figure as requested.

Could the authors include a section in the methods outlining what software they used to perform statistics.

>> All statistical analysis was performed in GraphPad Prism. The methods section has been revised to clarify this.

In Figure 5F line 860, the authors state that N=3, however it appears there are 6 replicates in that panel.

>> We have revised the figure legend accordingly.

In methods HtrA1 Δ NTD is mentioned as well as the protease-inactive mutant, but they don't

appear in manuscript. If this construct is not used, could the authors remove it from the methods section.

>> To clarify, all of our pure protein experiments were performed with the Δ NTD construct. We now have added the Δ NTD plasmid construct as well, which is used in the cell culture experiments. We have revised the text of the methods section to clarify.

References:

Cabrera, A. C., E. Melo, D. Roth, A. Topp, F. Delobel, C. Stucki, C.-y. Chen, P. Jakob, B. Banfai, T. Dunkley, O. Schilling, S. Huber, R. Iacone and P. Petrone (2017). "HtrA1 activation is driven by an allosteric mechanism of inter-monomer communication." Scientific Reports **7**(1): 14804.

Conway, K. A., S. J. Lee, J. C. Rochet, T. T. Ding, R. E. Williamson and P. T. Lansbury, Jr. (2000). "Acceleration of oligomerization, not fibrillization, is a shared property of both alpha-synuclein mutations linked to early-onset Parkinson's disease: implications for pathogenesis and therapy." Proc Natl Acad Sci U S A **97**(2): 571-576.

Flagmeier, P., G. Meisl, M. Vendruscolo, T. P. Knowles, C. M. Dobson, A. K. Buell and C. Galvagnion (2016). "Mutations associated with familial Parkinson's disease alter the initiation and amplification steps of α -synuclein aggregation." Proc Natl Acad Sci U S A **113**(37): 10328-10333.

Globus, O., T. Evron, M. Caspi, R. Siman-Tov and R. Rosin-Arbesfeld (2017). "High-Temperature Requirement A1 (Htra1) - A Novel Regulator of Canonical Wnt Signaling." Scientific Reports **7**(1): 17995.

Reardon, S. (2023). "FDA approves Alzheimer's drug lecanemab amid safety concerns." Nature **613**(7943): 227-228.

Rey, J., M. Breiden, V. Lux, A. Bluemke, M. Steindel, K. Ripkens, B. Möllers, K. Bravo Rodriguez, P. Boisguerin, R. Volkmer, J. Mieres-Perez, T. Clausen, E. Sanchez-Garcia and M. Ehrmann (2022). "An allosteric HTRA1-calpain 2 complex with restricted activation profile." Proc Natl Acad Sci U S A **119**(14): e21113520119.

Sevigny, J., P. Chiao, T. Bussière, P. H. Weinreb, L. Williams, M. Maier, R. Dunstan, S. Salloway, T. Chen, Y. Ling, J. O'Gorman, F. Qian, M. Arastu, M. Li, S. Chollate, M. S. Brennan, O. Quintero-Monzon, R. H. Scannevin, H. M. Arnold, T. Engber, K. Rhodes, J. Ferrero, Y. Hang, A. Mikulskis, J. Grimm, C. Hock, R. M. Nitsch and A. Sandrock (2016). "The antibody aducanumab reduces A β plaques in Alzheimer's disease." Nature **537**(7618): 50-56.

van Dyck, C. H., C. J. Swanson, P. Aisen, R. J. Bateman, C. Chen, M. Gee, M. Kanekiyo, D. Li, L. Reyderman, S. Cohen, L. Froelich, S. Katayama, M. Sabbagh, B. Vellas, D. Watson, S. Dhadda, M. Irizarry, L. D. Kramer and T. Iwatsubo (2022). "Lecanemab in Early Alzheimer's Disease." New England Journal of Medicine **388**(1): 9-21.

Reviewers' Comments:

Reviewer #1:

Remarks to the Author:

The authors have clarified several of the points that I made previously. However, the main conclusion is still not convincing.

They argue that HTRA1 binds α -synuclein fibrils, thereby promoting disaggregation. This conclusion is illustrated in Figure 7, at page 18: "To enable cleavage even in the fibrillar state, the protease domain of HTRA1 directly engages this aggregation-prone region of α -syn to mediate disaggregation, thereby allowing proteolytic cleavage to proceed (Fig 7)."

However, it is unclear how binding can ever be destabilizing. A bound state is always more stable than a free state, otherwise binding would not occur. If the protease domain of HTRA1 binds α -synuclein fibrils, the complex should be more stable than the free fibrils, otherwise the binding would not occur. However, there could still be disaggregation if the protease domain of HTRA1 binds even more strongly the monomeric state of α -synuclein.

The text in the manuscript is somewhat ambiguous throughout about the binding of HTRA1 to α -synuclein, since it does not systematically specify which state of α -synuclein is actually bound in the various experiments.

In short, the mechanism of action shown in Figure 7 is unlikely to be correct.

An experiment that the authors could perform is to measure the dissociation constants of the protease domain of HTRA1 with α -synuclein monomers and fibrils, respectively. The comparison between these two dissociation constants could provide convincing evidence for the proposed conclusion.

Reviewer #2:

Remarks to the Author:

My concerns were addressed well. Thank you.

Reviewer #3:

Remarks to the Author:

The revised manuscript is markedly improved. The authors were incredibly responsive and addressed the major concerns with this reviewer. No additional concerns were noted. This is an exciting report.

Reviewer #1:

The authors have clarified several of the points that I made previously. However, the main conclusion is still not convincing.

>> We thank the reviewer for recognizing the improvements to our manuscript. Based on the specific suggestions of the reviewer described below, we have performed new experiments and present additional evidence to support our model. We believe that our conclusions are now well-supported, as we describe below. We also have carefully worked through the manuscript to clarify elements of our wording and adjusted our mechanistic conclusions based on the new data. Finally, we describe the relevant limitations of our study and discuss outstanding questions that we do not resolve in this study.

They argue that HTRA1 binds α -synuclein fibrils, thereby promoting disaggregation. This conclusion is illustrated in Figure 7, at page 18: "To enable cleavage even in the fibrillar state, the protease domain of HTRA1 directly engages this aggregation-prone region of α -syn to mediate disaggregation, thereby allowing proteolytic cleavage to proceed (Fig 7)."

However, it is unclear how binding can ever be destabilizing. A bound state is always more stable than a free state, otherwise binding would not occur. If the protease domain of HTRA1 binds α -synuclein fibrils, the complex should be more stable than the free fibrils, otherwise the binding would not occur. However, there could still be disaggregation if the protease domain of HTRA binds even more strongly the monomeric state of α -synuclein.

>> This is an important point. We agree with the reviewer that a bound state is typically more stable than a free state. However, binding and consequent disaggregation is not necessarily equivalent to destabilization. Rather, we propose that the mechanism of HTRA1-mediated disaggregation is similar to the mechanisms of other ATP-independent disaggregases, such as DAXX (Huang et al., *Nature*, 2021) and Kap β 2 (Guo et al., *Cell*, 2018). In the DAXX study, the authors demonstrated that the disaggregation mechanism might be mediated by electrostatic binding interactions. In the Kap β 2 study, Kap β 2 was found to initially bind the PY-NLS region of FUS to disaggregate fibrils. Further, they proposed that the binding of Kap β 2 to the PY-NLS may enable disaggregation via entropic pulling, and the binding possibly drives a long-range allosteric conformational change that breaks cross-beta fibril contacts. Additionally, earlier studies investigated the mechanism of the ATP-independent protein disaggregase cpSRP43 (Jaru-Ampornpan et al., *J. Biol. Chem*, 2013; Jaru-Ampornpan et al., *Nat Struct Mol Biol.*, 2010). Here, the authors showed that disaggregation can be mediated by specific binding interactions and recognition on the aggregate surface, not the monomer. In an early study of HTRA1, Poepsel et al. showed that HTRA1:S328A binds to the tau fibril surface, again not the monomer, and this binding mediates disintegration of tau fibrils (Poepsel et al., *Nat Chem Biol*, 2015). Further, a nanobody was developed and described in a study from Butler et al. Here, the nanobody is shown to specifically recognize and bind α -synuclein fibrils, but not monomer. The authors go on to demonstrate that this nanobody has ATP-independent disaggregation

activity against α -synuclein preformed fibrils (Butler et al., *Nat Commun.*, 2022). In sum, while we acknowledge that our understanding of this mechanism is not fully complete, we note well-established evidence that *ATP-independent disaggregases can function via a binding-based mechanism*, and that *binding to the fibrillar species rather than the monomer is capable of driving disaggregation*.

Nonetheless, it is important to also consider the possibility that HTRA1 could function via an entirely different mechanism than the disaggregases described above. Our understanding is that the reviewer is proposing a mechanism whereby disaggregation by HTRA1 is mediated by binding to monomeric α -synuclein. From the reviewer's comments, we think they are suggesting a model whereby HTRA1 binds monomeric α -synuclein and acts on it, and then as monomer is depleted, a monomer-fibril equilibrium is disrupted which leads to restoration of equilibrium via dissociation of fibrils and accumulation of additional monomer. This free monomer would then be bound and acted on by HTRA1. If our interpretation of the reviewer's suggestion is correct, we see several problems with this model:

First, such a model might explain *inhibition* of α -synuclein aggregation but would not explain *disaggregation*. If HTRA1 more favorably binds the monomeric form of α -synuclein, then α -synuclein fibrils would be resistant to dissolution. However, in Figures 5 and 6 we are treating pre-formed α -syn fibrils (PFFs) with HTRA1 (in a system with little, if any, monomer present) and we see strong clearance of these fibrillar materials, suggesting that HTRA1 is engaging the fibrillar form of α -synuclein rather than the monomer. Engagement and activity against the monomer alone would only prevent aggregation, but not reverse it. In Figure 5, we generate α -syn PFFs and demonstrate that HTRA1SA treatment drives clearance of ~40% of the fibrillar material. These fibrillar materials are generated by shaking for 7 days. While we cannot confirm that no monomer remains in solution after 7 days of shaking, based on our sedimentation assays shown in Figure 2C, we see that nearly 100% of α -synuclein shifts to the insoluble (fibrillar) fraction after just 72h of shaking (see buffer or aldolase treated lanes). By extrapolation, and in the absence of fibrillar dissociation to monomer, the most plausible interpretation of the results shown in Figure 5 is that HTRA1 acts primarily on the fibrillar species.

Second, in the absence of a disaggregase, amyloid fibril formation is an irreversible process and there is no appreciable monomer – fibril equilibrium. Monomeric amyloid species are known to undergo nucleation followed by fibrillization. These fibrils are highly stable and only minimal dissociation to monomer occurs. In Parkinson's disease as well, α -synuclein fibrils are resistant to clearance, and result in accumulation of insoluble deposits (Forman et al., *Nat Med*, 2004; Lashuel et al., *Nat Rev Neurosci*, 2013; Tanik et al., *J. Biol. Chem.*, 2013). It is important to note that the amyloid aggregation process is distinct from typical chemical reactions, in that there is no appreciable equilibrium or reversibility, rate-determining step, etc. It is also important to note that these assumptions regarding lack of spontaneous reversibility do not account for the presence of a disaggregase.

Third, if this model were true, where binding of monomer alone is sufficient to mediate disaggregation, then we would expect that *any protein* that binds α -synuclein tightly, such as an antibody, would mediate disaggregation. From the large number of studies employing α -synuclein antibodies, which have not been shown to mediate disaggregation, it is apparent that binding alone is insufficient to drive disaggregation. Furthermore, in the Kap β 2 studies (Guo et al., *Cell*, 2018), the authors showed that an antibody specific to the PY-NLS of FUS does not disaggregate FUS aggregates. Similarly, *in our study*, we

discovered that HTRA2 binds α -synuclein monomer (Fig 4G), yet it does not inhibit α -synuclein aggregation. Additional evidence that binding to fibrils can mediate disaggregation comes from the study by Butler et al., where they demonstrate that this nanobody has disaggregation activity against α -synuclein preformed fibrils, and the nanobody does not bind monomer (Butler et al., *Nat Commun.*, 2022).

Alternatively, another possibility that we thought the reviewer may be suggesting is that HTRA1 is not acting on the fibrillar species per se, but instead favorably binding to the terminal ends of fibrils, where these α -syn subunits may more closely resemble the monomeric state. In this scenario, HTRA1 may then convert the conformation of these terminal α -synuclein molecules to a state that is no longer able to template with the fibril. However, our newly provided data (Figure 6F-G below) do not support this model either, as we note tighter binding to the fibrillar form of α -synuclein. We also would not want to speculate on the specific region on the elongated fibrillar surface that HTRA1 binds to. While this is an important future direction, we believe this is beyond the scope of our present study.

In summary, we respectfully disagree with the reviewer that disaggregation can only occur if HTRA1 binds more strongly to the monomeric rather than fibrillar state of α -synuclein. Rather, our interpretation is that disaggregases can function in an ATP-independent fashion and mediate disaggregation by binding to the fibrillar species.

The text in the manuscript is somewhat ambiguous throughout about the binding of HTRA1 to alpha-synuclein, since it does not systematically specify which state of alpha-synuclein is actually bound in the various experiments.

>> We thank the reviewer for pointing this out. We had tried to clarify which state of α -synuclein is being used in each of the experiments in our first revision. Now with better understanding of the reviewer's concern, we have gone back through and added further description at several key points. If there are any additional areas where it is unclear which species is being used we are happy to add additional clarification. We agree that such descriptions are essential for accurate interpretation and we hope that the manuscript reads more clearly now.

We also wanted to highlight that we are not claiming that HTRA1 engages α -syn fibrils and returns the protein to a monomeric state. We have not been able to characterize the remodeled species with the resolution needed to determine this. Instead, we are proposing that HTRA1 remodels the fibrils to a species that is no longer toxic or seeding competent, but this species may not necessarily be monomeric.

In short, the mechanism of action shown in Figure 7 is unlikely to be correct. An experiment that the authors could perform is to measure the dissociation constants of the protease domain of HTRA1 with alpha-synuclein monomers and fibrils, respectively. The comparison between these two dissociation constants could provide convincing evidence for the proposed conclusion.

>> We thank the reviewer for carefully considering the strength of our model, and it is our objective to ensure that our model fully aligns with the experimental evidence that we have acquired. First, it is important to note that we do not believe that the current manuscript provides a comprehensive elucidation of all aspects of the mechanism of HTRA1-mediated disaggregation. Instead, our objective is to demonstrate that HTRA1 dissolves α -synuclein PFFs, and that this dissolution detoxifies the PFFs and renders them seeding-incompetent. We suggest that a complete understanding of this mechanism is beyond the scope of the present study. Full understanding of this mechanism will require several additional years of investigation. We also are already presenting nine main figures and eight supplemental figures. Thus we believe that adding extensive additional data to support our mechanism would make the manuscript too broad and cumbersome for the reader. However, we fully agree with the reviewer that additional mechanistic understanding would be interesting and impactful, and we intend to focus on this in the coming years.

For the present study, we agree with the reviewer that their proposed experiments to measure dissociation constants would be worthwhile to pursue and would increase the impact of our study. However, it is important to note that based on our interpretation of our data and the literature, in initiating these experiments, we suggest that the opposite result would better support our proposed model. Specifically, we predicted that HTRA1 should bind fibril more tightly than monomer.

The reasons for this are explained above on pages 2-3. This is what we ultimately observed, and we believe that this result is fully consistent with our proposed model.

Here, we performed a pull-down assay to compare the relative binding affinity of the protease domain of HTRA1 with α -synuclein monomers and fibrils. This new data is shown in Figure 6F-G. We have performed four biological replicates of this experiment (all four trials are shown in the source data file for reference), and see a statistically significant tighter binding to α -synuclein PFFs. We acknowledge that the reviewer had requested that we measure and compare the dissociation constants of HTRA1 to α -synuclein monomer and fibril. We note that pull-down assays do not quantitatively measure dissociation constants, and we do not want to over-interpret these experiments by attempting to extrapolate dissociation constants from such assays. Nonetheless, we feel that we can make semi-quantitative conclusions from these assays.

Quantitative measurement of the dissociation constants towards α -synuclein PFFs has proven particularly challenging. While we were able to optimize the fluorescence polarization binding assays shown in Figure 4I in our manuscript, these assays used α -synuclein monomer. We have attempted to optimize this assay using α -synuclein PFFs as well as a quantitative ELISA assay. In the polarization assays, we observed that the labeled PFFs produce a high background signal, presumably due to their higher-order structure, poor solubility, and slow tumbling rate, which prevented us from obtaining a reliable working signal. We next attempted to optimize this assay in a competition format. Here, we incubated the fluorescently-labeled monomeric protein with HTRA1ProD^{SA} and titrated in unlabeled α -synuclein PFFs which could serve as inhibitors of the binding interaction. Here, we observed an increase in the polarization signal upon incubation. We believe this is due to the fibrils binding the monomer to continue the process of amyloidogenesis. We also attempted ELISA assays, where we tried to optimize a published protocol (Butler et al., *Nat Commun.*, 2022). Here, we planned to coat a plate with α -synuclein monomer and PFFs and then titrate HTRA1

Figure 6F-G. Newly added pull-down data. (F) His-HTRA1-ProD^{SA} (10 μ M) was immobilized on Ni-NTA resin and incubated with α -syn monomer or pre-formed fibrils (PFFs) (20 μ M) overnight. Beads were then washed and the input and bound fractions were processed by immunoblotting using an α -syn antibody. (G) Experiments from F were quantified and normalized to the α -syn monomer condition. Values are compared to control reactions with α -syn monomer using a one-way ANOVA with a Dunnett's multiple comparisons test (N = 4, biological replicates are shown as dots, bars represent means \pm SEM, * p < 0.05).

concentrations to determine the binding affinity of these interactions. Unfortunately, in just the nanomolar range, we observed high background signal and non-specific binding to the HTRA1 treated wells on the ELISA plate surface. We repeated this optimization using a number of different blocking buffers, but have not been able to identify an effective blocking buffer. Therefore, while we had hoped to be able to provide fully quantitative data, it seems that we will not be able to perform these experiments and instead we have decided to report the semi-quantitative pull-down assays.

This newly added pull-down assay data provides evidence that HTRA1 binds fibrils more tightly than monomer. This data has been added to the manuscript (Figure 6F-G). We have also carefully modified the text of the results and the discussion sections to ensure that our description of the mechanism is well-supported by our data. To summarize, we have made the following changes:

- Throughout the manuscript we have added additional descriptors to indicate which experiments were performed with monomer vs. PFFs.
- The newly added data is described in the results section and shown in Figure 6F-G.
- We have edited Figure 7 to better align with our data and minimize speculation. Specifically, while we have found that binding to fibrils is favored as compared to monomer, we do not know if HTRA1 binds the NAC domain to mediate disaggregation or if perhaps it binds to another region of α -syn and allosterically remodels the NAC domain. Therefore we have removed any indication of where HTRA1 binds for disaggregation, and only show it binding the NAC domain at the proteolysis step, where it must interact with the NAC to mediate cleavage.
- We have also modified the legend of Figure 7 to add additional descriptors to aid in interpretation of the model. If the reviewer still has reservations about this model, we are also willing to remove this figure entirely if that is preferred.
- The wording throughout the description of the model in the results section, and particularly the sentence in question quoted by the reviewer, has been modified carefully to tone down our conclusions and avoid overinterpretation of our data.
- In the discussion section we have carefully edited our description of the model to ensure all statements are supported by data and toned down our conclusions where appropriate. We have also added a discussion of the relevant limitations of our study to provide readers with a more complete understanding of our work.

In sum, we thank the reviewer for their rigorous assessment of our work. We believe that with these new experiments, we have improved the nuance of our mechanism which will increase the impact of our work.

Reviewer #2:

My concerns were addressed well. Thank you.

Reviewer #3:

The revised manuscript is markedly improved. The authors were incredibly responsive and addressed the major concerns with this reviewer. No additional concerns were noted. This is an exciting report.

References

Butler, Y.R., Liu, Y., Kumbhar, R., Zhao, P., Gadhave, K., Wang, N., Li, Y., Mao, X., and Wang, W. (2022). α -Synuclein fibril-specific nanobody reduces prion-like α -synuclein spreading in mice. *Nat Commun* 13, 4060.

Forman, M.S., Trojanowski, J.Q., and Lee, V.M. (2004). Neurodegenerative diseases: a decade of discoveries paves the way for therapeutic breakthroughs. *Nat Med* 10, 1055-1063.

Guo, L., Kim, H.J., Wang, H., Monaghan, J., Freyermuth, F., Sung, J.C., O'Donovan, K., Fare, C.M., Diaz, Z., Singh, N., *et al.* (2018). Nuclear-Import Receptors Reverse Aberrant Phase Transitions of RNA-Binding Proteins with Prion-like Domains. *Cell* 173, 677-692.e620.

Huang, L., Agrawal, T., Zhu, G., Yu, S., Tao, L., Lin, J., Marmorstein, R., Shorter, J., and Yang, X. (2021). DAXX represents a new type of protein-folding enabler. *Nature* 597, 132-137.

Jaru-Ampornpan, P., Liang, F.C., Nisthal, A., Nguyen, T.X., Wang, P., Shen, K., Mayo, S.L., and Shan, S.O. (2013). Mechanism of an ATP-independent protein disaggregase: II. distinct molecular interactions drive multiple steps during aggregate disassembly. *J. Biol. Chem.* 288, 13431-13445.

Jaru-Ampornpan, P., Shen, K., Lam, V.Q., Ali, M., Doniach, S., Jia, T.Z., and Shan, S.O. (2010). ATP-independent reversal of a membrane protein aggregate by a chloroplast SRP subunit. *Nat Struct Mol Biol* 17, 696-702.

Lashuel, H.A., Overk, C.R., Oueslati, A., and Masliah, E. (2013). The many faces of α -synuclein: from structure and toxicity to therapeutic target. *Nat. Rev. Neurosci.* 14, 38-48.

Poepsel, S., Sprengel, A., Sacca, B., Kaschani, F., Kaiser, M., Gatsogiannis, C., Raunser, S., Clausen, T., and Ehrmann, M. (2015). Determinants of amyloid fibril degradation by the PDZ protease HTRA1. *Nat Chem Biol* 11, 862-869.

Tanik, S.A., Schultheiss, C.E., Volpicelli-Daley, L.A., Brunden, K.R., and Lee, V.M. (2013). Lewy body-like α -synuclein aggregates resist degradation and impair macroautophagy. *J. Biol. Chem.* 288, 15194-15210.